# Monoclonal antibodies from humans with *Mycobacterium tuberculosis* exposure or latent infection recognize distinct arabinomannan epitopes

Elise Ishida [1], Devin T. Corrigan [2], Ryan J. Malonis[3], Daniel Hofmann[3], Tingting Chen[2], Anita G. Amin[4], Delphi Chatterjee[4], Maju Joe[5], Todd L. Lowary [5,6,7], Jonathan R. Lai[3] & Jacqueline M. Achkar [1,2✉]

The surface polysacharide arabinomannan (AM) and related glycolipid lipoarabinomannan (LAM) play critical roles in tuberculosis pathogenesis. Human antibody responses to AM/LAM are heterogenous and knowledge of reactivity to specific glycan epitopes at the monoclonal level is limited, especially in individuals who can control *M. tuberculosis* infection. We generated human IgG mAbs to AM/LAM from B cells of two asymptomatic individuals exposed to or latently infected with *M. tuberculosis*. Here, we show that two of these mAbs have high affinity to AM/LAM, are non-competing, and recognize different glycan epitopes distinct from other anti-AM/LAM mAbs reported. Both mAbs recognize virulent *M. tuberculosis* and nontuberculous mycobacteria with marked differences, can be used for the detection of urinary LAM, and can detect *M. tuberculosis* and LAM in infected lungs. These mAbs enhance our understanding of the spectrum of antibodies to AM/LAM epitopes in humans and are valuable for tuberculosis diagnostic and research applications.

[1] Department of Microbiology and Immunology, Albert Einstein College of Medicine, Bronx, NY, USA. [2] Department of Medicine, Albert Einstein College of Medicine, Bronx, NY, USA. [3] Department of Biochemistry, Albert Einstein College of Medicine, Bronx, NY, USA. [4] Mycobacteria Research Laboratories, Department of Microbiology, Immunology and Pathology, Colorado State University, Fort Collins, CO, USA. [5] Department of Chemistry, University of Alberta, Edmonton, AB, Canada. [6] Institute of Biological Chemistry, Academia Sinica, Nangang, Taipei, Taiwan. [7] Institute of Biochemical Sciences, National Taiwan University, Taipei, Taiwan. ✉email: jacqueline.achkar@einsteinmed.org

With over 10 million cases per year and one million associated deaths, active tuberculosis (TB), caused by the facultative intracellular pathogen *Mycobacterium tuberculosis* (*Mtb*), is, next to COVID-19, the leading cause of death from a single infectious agent[1]. While an estimated quarter of the world is latently infected with *Mtb*[2], TB is caused by uncontrolled infection leading to a predominantly respiratory and transmissible disease. To combat this major global public health problem, better vaccines, therapies, and additional tools for both diagnosis and research are critical. Beyond their potential to inform vaccine and immunotherapy development, antibodies are versatile and indispensable tools in a plethora of applications in medicine and research, including the detection of pathogens and their antigens. For example, there is a resurgence of interest in the generation of monoclonal antibodies (mAbs) to the mycobacterial cell wall glycolipid lipoarabinomannan (LAM) to improve the limited sensitivity of currently available urinary LAM (U-LAM) detection tests for the diagnosis of TB (reviewed in Shah et al.[3])

For both diagnostic purposes and the investigation of protective efficacy, we are interested in exploring the breadth and functions of human antibodies to *Mtb*, especially those to the *Mtb* surface. The mycobacterial capsule is an important virulence factor that is predominantly made up of proteins and two polysaccharides: α-glucan and arabinomannan (AM)[4–6]. AM and its structurally related cell wall and membrane glycolipid LAM play a critical role in TB pathogenesis (reviewed in Angala et al.[7], Turner & Torrelles[8], Kalscheuer et al.[9], Correia-Neves et al.[10]) and have been a focus of our investigations. We have shown that human serum IgG titers to AM and LAM strongly correlate[11,12], and that anti-AM polyclonal murine and human antibodies have protective functions against *Mtb* in vitro and in vivo[13–15]. Using synthetic glycan arrays, we further demonstrated that human serum immunoglobulin G (IgG) antibodies induced by BCG vaccination and natural *Mtb* infection, are tremendously heterogeneous in their binding specificity to AM oligosaccharides[13,15]. Our data showed that anti-AM polyclonal IgG from individuals with BCG vaccination and/or latent *Mtb* infection (LTBI) have protective functions, and further suggested that targeting specific glycan epitopes within AM could be relevant for protective efficacy[13,15].

Polysaccharide antigens have a higher degree of flexibility than proteins[16]. To better understand and study the spectrum of human antibodies targeting different AM epitopes at the monoclonal level, especially in individuals who can control *Mtb* infection, our objective was to generate and characterize human mAbs to *Mtb* capsular AM. To date, few human mAbs to *Mtb* antigens have been generated. In two recent studies, recombinant human mAbs to LAM were created through arduous approaches—sorting and cloning all circulating plasmablasts and/or sorting, culturing, stimulating, and screening all memory B cells from patients with TB[17,18]. Among these, the best characterized mAb A194 has high affinity and recognizes a range of AM oligosaccharide motifs sharing the uncapped Ara4 and Ara6 epitopes commonly recognized by murine mAbs[18]. We here used AM-specific single B cell sorting to develop a targeted approach for generating mAbs from asymptomatic BCG vaccinated and *Mtb* exposed or infected individuals. The sera from these individuals had high anti-AM IgG titers, protective polyclonal anti-AM IgG functions, and diverse glycan epitope binding[15]. Our hypotheses were: (1) Single B cell sorting with *Mtb* capsular AM as a probe could be used to generate human mAbs to a spectrum of AM epitopes; and (2) Among the generated mAbs to AM, those with high affinity could have value as TB diagnostic and research tools. Here, we describe an efficient strategy to generate diverse human mAbs to AM and show that those with high affinity recognize different glycan epitopes, distinct from other anti-AM/LAM mAbs reported to date. These proteins

represent valuable tools for the TB field in both diagnostic and research applications and enhance our knowledge of antibody epitope specificity to *Mtb* glycans in humans.

## Results

**Generation of human mAbs to the *Mtb* capsular polysaccharide AM**. To characterize and better understand the spectrum of anti-AM serum IgG responses in subjects exposed to or infected with *Mtb*, we generated human mAbs using a targeted, flow cytometry-based strategy[19–21]. In our recent studies, subjects from a range of *Mtb* exposure and infection along the clinical spectrum were tested for their serum anti-AM IgG responses[15]. Subjects T1 and L1 were selected for single B cell sorting and antibody generation based on availability of peripheral blood mononuclear cells (PBMC), high serum anti-AM IgG titers, and protective polyclonal anti-AM IgG functions against *Mtb*[15]. Both donors had emigrated from TB endemic regions to the US during childhood or adolescence, had been BCG vaccinated, were asymptomatic, had never received TB preventative therapy, and had no chest X-ray abnormalities. Subject T1, who during early childhood had a close family member die of presumed TB, was tuberculin skin test (TST) positive (>15 mm) and whole blood interferon-gamma release assay (IGRA; Quantiferon TB Gold) negative. Subject L1 reported no known TB exposure and was TST positive (>15 mm) and IGRA positive. We obtained serial sera from both donors over several years and found persistently high anti-AM IgG titers (Supplementary Fig. 1).

We sorted for AM-positive, IgG-positive B cells using two sorting strategies (Fig. 1a, b, Supplementary Fig. 2). For our first sort with PBMCs from subject T1, we sorted for CD20+, CD27+ antigen-positive B cells[21,22]. Of the around 400 AM-positive B cells (CD20+, CD27+) from subject T1, 17.9% (73 single cells) were IgG+ and sorted into individual wells (Fig. 1a). To potentially increase the yield and quality of AM + IgG+ single B cells when sorting cells from subject L1, we applied a second strategy consistent with other reports that sorted B cells to oligosaccharides from other pathogens[23,24]. Of the around 675 AM-positive B cells (CD19+) from subject L1, 9.6% (65 single cells) were IgG+ and sorted into individual wells (Fig. 1b). When repeating this second sorting strategy with T1 PBMCs, we neither increased our yield nor identified additional mAbs to AM (Supplementary Fig. 3). About 12% of AM and IgG-positive B cells expressed adequate quantities of protein. This was lower than the 25–40% efficiency reported in other mAb generation studies, but PBMC from these studies were isolated from recently vaccinated or symptomatic individuals and were sorted using probes for protein antigens[25,26]. Antibody sequence repertoire analysis revealed that mAbs came from distinct germline lineages (Supplementary Tables 1 and 2).

**Binding characteristics of human mAbs to AM**. Consistent with antibodies to polysaccharide antigens[27–29], we found that most of the mAbs were <10% mutated from the germline sequence (Supplementary Tables 1, 2) and required high concentrations to detect reactivity to capsular AM, suggesting low binding affinities (Fig. 1c, d). One mAb from each subject, T1AM09 and L1AM04, showed much stronger binding to AM by ELISA than the other mAbs; they were thus chosen for in-depth characterization and assessment of TB diagnostic and research value.

**Human high-affinity anti-AM mAbs have distinct binding characteristics and reactivities to glycan epitopes**. Both T1AM09 and L1AM04 bound the strongest to *Mtb* capsular AM, followed by cell wall LAM (Fig. 2a, b). Their weaker binding to the *Mtb* cell wall, membrane, and culture filtrate fractions were

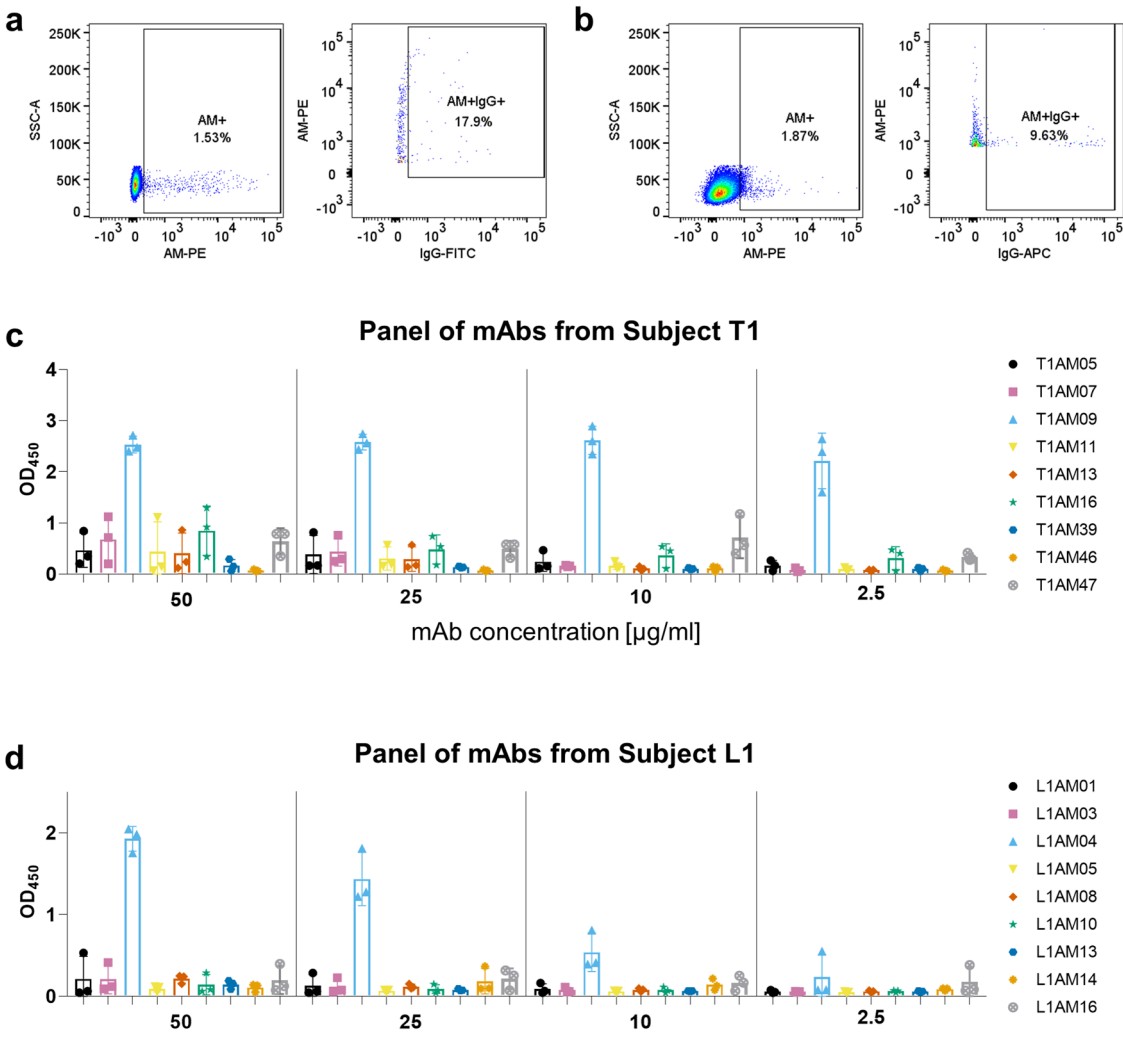

**Fig. 1 Human B cells and monoclonal antibodies (mAbs) bind with ranging affinities to arabinomannan (AM).** Binding of human B cells to AM by flow cytometry for **a** subject T1 (CD20+, CD27+, IgG+), **b** subject L1 (CD19+, IgG+). Binding of mAbs to H37Rv AM by ELISA at varying concentrations from **c** subject T1, and **d** subject L1. Florescence-minus-one (FMO) controls were used to set conservative AM+ gates and IgG+ gates for sorting. Each bar represents the mean and standard deviation from three experiments.

expected due to the lower AM/LAM concentrations in these fractions compared to purified AM/LAM. The differences in binding between L1AM04 and T1AM09 to LAM and these fractions suggested differences in AM epitope binding and/or affinity. L1AM04 scarcely bound to the *Mtb* cell wall glycolipid lipomannan (LM), and overall both mAbs had limited binding to this antigen (Fig. 2a, b). Because LM compared to LAM lacks the arabinan domain (reviewed in Turner and Torrelles[8]), these data suggest that the mAbs recognize the arabinan component of AM/LAM and/or a mannose configuration absent in LM. Both mAbs had comparable binding to the biotinylated H37Rv probe (used for sorting) and AM from virulent strains of *Mtb*, indicating that the probe used to capture anti-AM-specific B cells resembles that of native AM (Fig. 2c, d). Relative to the glycans isolated from virulent *Mtb* strains, L1AM04 bound less to AM isolated from BCG and H37Ra, suggesting that this mAb can capture important differences in AM glycan epitopes between *Mtb* and mycobacterial species with reduced virulence (Fig. 2d; Supplementary Table 3).

To determine the AM epitopes recognized by T1AM09 and L1AM04, we tested mAb binding to 63 synthetically generated oligosaccharides from mycobacterial surface glycans which, in addition to the 30 fragments from AM and LAM, included fragments from six other glycan classes (α-glucan, trehalose mycolates, lipooligosachharides (LOSs), phenolic glycolipids (PGLs), phosphatidyl-*myo*-inositol mannosides (PIMs) and glycopepitdolipids (GPLs)[13,30]. T1AM09 had the highest reactivity to the AM fragments S19, S21, and S22, arabinan motifs sharing long chains of arabinose residues, and did not require the Ara-β(1 → 2)-Ara linkage at the terminal nonreducing position for binding (Fig. 3a, c). L1AM04 reacted with AM fragments S4, S9, and S25, sharing a specific trimannoside capped arabinan motif, $Man_3Ara_4$, from the group of *Mtb* mannose-capped LAM (ManLAM; Fig. 3b, c). The reactivity of L1AM04 with one PIM oligosaccharide (S23, also referred to as $PIM_6$) is likely based on cross-reactivity to the shared three-mannose residue motif.

**Binding kinetics of anti-AM mAbs.** We further investigated the binding of T1AM09 and L1AM04 to AM by Biolayer Interferometry (BLI). Biotinylated AM was immobilized onto streptavidin sensors and then dipped into solutions of T1AM09 or L1AM04 at various concentrations. Given the bivalent nature of

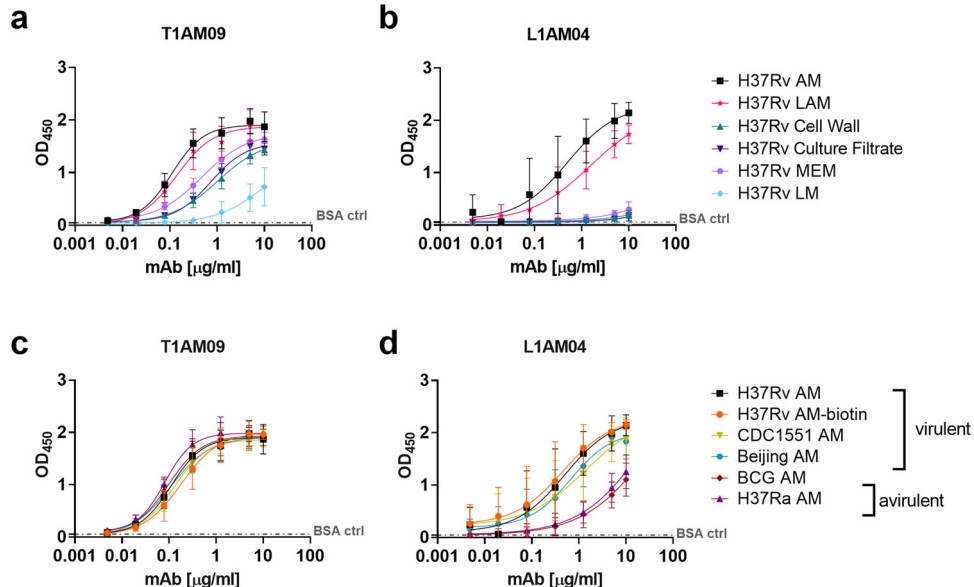

**Fig. 2 Anti-AM cross-reactivity of T1AM09 and L1AM04 to *Mtb* components.** Binding of mAbs assessed by ELISA to antigens and fractions isolated from H37Rv for **a** T1AM09, and **b** L1AM04, and to AM isolated from different *Mtb* strains for **c** T1AM09, and **d** L1AM04. Each data point represents the mean and standard deviation from two or more replicates. Dashed line represents the BSA control.

the IgGs, this measurement does not provide an unambiguous determination of binding kinetics without the potential for avidity but nonetheless can afford insight into comparative affinities. Hence, the reported dissociation constants are referred to as apparent $K_D$ ($K_{D, app}$). Both mAbs bound to AM with $K_{D, app}$ values in the sub-micromolar range and rapid association rates (Table 1, Fig. 4a). We further compared the binding kinetics of T1AM09 and L1AM04 to the well-known anti-LAM murine mAb CS-35, which recognizes most AM oligosaccharide motifs within AM[31–33] (Supplementary Fig. 4). Two-phase binding experiments by BLI suggested these anti-AM/LAM mAbs do not compete for binding to AM (Fig. 4c, d, Supplementary Fig. 4). Briefly, to assess epitope binding, biotinylated AM was first immobilized on a streptavidin sensor and then allowed to equilibrate in a solution containing the first mAb. After the binding plateaus, the biosensor was dipped into a second well containing equimolar amounts of the first mAb and test mAb to determine whether the mAbs bind distinct or shared epitopes (Fig. 4c, d). For both T1AM09 or L1AM04, the observed second association curve suggests that these mAbs bind different AM epitopes. Furthermore, T1AM09 does not target the CS-35 epitope (Supplementary Fig. 4).

**Potential applications of T1AM09 and L1AM04 in diagnostics and imaging.** The detection of LAM in the urine of TB patients is based on the use of capture and detection mAbs recognizing different LAM epitopes[34,35]. To assess whether the distinct AM epitope specificity of T1AM09 and L1AM04 (Figs. 3 and 4) could have value in a U-LAM sandwich ELISA, we characterized the binding of T1AM09 and L1AM04 to serially diluted LAM spiked into urine from a healthy volunteer (Fig. 5a). Due to its improved detection of U-LAM compared to commercially available tests, the mAb pair CS-35 and A194 was used as a reference (Fig. 5b)[34,36]. Using L1AM04 as the capture mAb and biotinylated T1AM09 as the detection mAb was highly sensitive for detecting the spiked LAM (limit of detection ~20 pg/mL). Moreover, the T1AM09/L1AM04 pair also detected LAM in clinical urine samples comparable to the CS-35/A194 pair and distinguished two HIV uninfected patients with

pulmonary TB from two patients with other respiratory diseases than TB (Table 2). These data suggest that including high-affinity human anti-AM mAbs generated through single B cell sorting and recognizing distinct glycan epitopes might contribute to the improvement of currently available U-LAM detection tests.

Using fluorescence microscopy, we showed that both mAbs T1AM09 and L1AM04 (10 μg/mL) bind strongly to the capsule of virulent laboratory (H37Rv and Erdman) and clinical strains (CDC1551 and Beijing) of the *Mtb* complex group when compared to positive and negative controls (Fig. 6 and Supplementary Fig. 5). Of the avirulent strains, both mAbs exhibited weaker binding to the *Mtb* mutant H37Ra and bound to BCG with varying degrees of intensity, suggesting diversity of the mAb-specific glycan epitopes or their accessibility in these strains (Fig. 6a, b). Of the nontuberculous mycobacteria, T1AM09 bound to M. abscessus to a lesser degree than to virulent *Mtb* strains, but not to M. avium, and L1AM04 bound to neither, supporting differences in the mAb-specific glycan epitopes or their accessibility in these strains (Fig. 6a, b).

We quantified the binding of the mAbs to different mycobacterial strains by a whole-cell ELISA. When quantifying the mAb binding by a whole-cell ELISA, T1AM09 bound to virulent *Mtb* strains and avirulent strains of the *Mtb* complex group at higher optical densities (ODs) than nontuberculosis strains (NTM), M. avium and M. abcessus (Fig. 7a). L1AM04 bound to virulent *Mtb* strains H37Rv, Erdman, and CDC1551 at higher ODs compared to Beijing and BCG (Fig. 7b), suggesting that these strains have different quantities of Man3 exposed on the whole bacteria. L1AM04 neither bound to H37Ra nor the NTM strains at a higher OD compared to the isotype negative control. While T1AM09 had weak binding and L1AM04 did not bind to the NTM strains tested, serum from both subjects these mAbs originated from (T1 and L1) showed strong binding to M. avium and M. abcessus, demonstrating that whole-cell ELISA can detect similar binding of polyclonal sera to these NTM and virulent *Mtb* strains (Fig. 7c–e).

To investigate the value of T1AM09 and L1AM04 for the detection of *Mtb* and AM/LAM in the lung tissue of *Mtb*

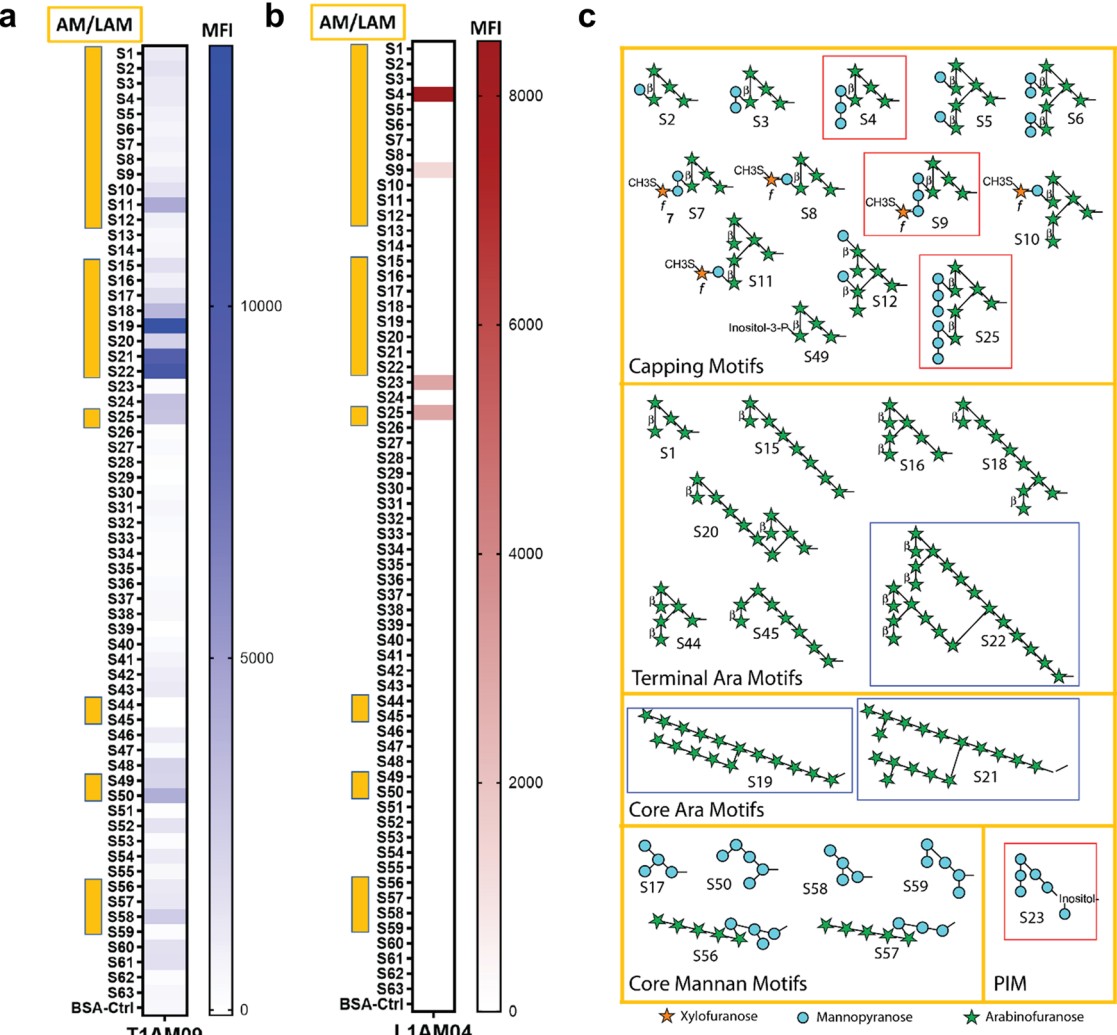

**Fig. 3 Reactivity of mAbs T1AM09 and L1AM04 to mycobacterial envelope glycan epitopes.** Median Fluorescence Intensity (MFI) of **a** T1AM09 (5 μg/mL), and **b** L1AM04 (5 μg/mL) binding 63 mycobacterial oligosaccharide fragments. Arabinomannan/lipoarabinomannan (AM/LAM) specific fragments (S#1–12, 15–22, 25, 44, 45, 49, 50, 56–59) are marked by the yellow side bar. Six other glycans on the array are: α-glucan (S#13, 14, 24, 46, 48, 52), trehalose mycolates and lipooligosachharides (LOSs; S#38, 39, 54, 55), phenolic glycolipids (PGLs; S#26–37, 40–43, 50, 53), phosphatidyl-myo-inositol mannosides (PIMs, S#23) and glycopepitdolipids (GPLs; S#47, 60, 61). **c** AM/LAM and PIM motifs with those most strongly recognized by T1AM09 (blue) and L1AM04 (red). Each data point (S1–S63) represents the mean from two or more replicates. All bonds are alpha unless designated in the figure as beta (β).

| Table 1 Binding kinetics of mAbs analyzed by Bio Layer Interferometry. | | | | | | |
|---|---|---|---|---|---|---|
| Antibody ID | $K_{D,app}$ (M) | $K_{D,app}$ Error | $k_{on}$ (1/Ms) | $k_{on}$ Error | $k_{dis}$ (1/s) | $k_{dis}$ Error |
| T1AM09 | 2.6E−08 | 3.5E−10 | 2.6E+05 | 3.1E+03 | 6.9E−03 | 4.0E−05 |
| L1AM04 | 9.6E−08 | 5.6E−09 | 2.5E+05 | 1.8E+04 | 1.9E−02 | 3.3E−04 |

CDC1551 and Erdman infected mice, we evaluated mAbs by immunohistochemistry (IHC) staining using Acid-Fast Bacilli (AFB) staining (Ziehl–Neelsen) as a positive control. Both mAbs detected extracellular and intracellular bacilli and AM/LAM throughout infected lungs with minimal off-target effects (Fig. 8). By contrast, AFB staining showed weaker positivity for *Mtb* CDC1551 bacilli (Fig. 8c), was negative for Erdman bacilli (Fig. 8f), and did not improve when the alternative Fite's method was used (Supplementary Fig. 6). These data show the value of T1AM09 and L1AM04 for the in situ detection of LAM and suggest that both mAbs could improve the sensitivity for *Mtb* detection in infected tissue.

## Discussion

We show that human mAbs, generated by single B cell sorting to AM with PBMC from two asymptomatic *Mtb* exposed or latently infected individuals, recognize distinct AM/LAM glycan epitopes. The two mAbs, T1AM09 and L1AM04, have, for anti-glycan mAbs, high affinity to *Mtb* capsular AM and its related cell wall glycolipid LAM. Compared to the polyclonal serum IgG, the high-affinity mAbs recognize only a few AM epitopes, distinct from other reported anti-AM/LAM mAbs, thereby enhancing our knowledge of antibody epitope specificity to mycobacterial gly- cans in humans. While T1AM09 recognizes predominantly the arabinan domain, L1AM04 recognizes a trimannoside-capped

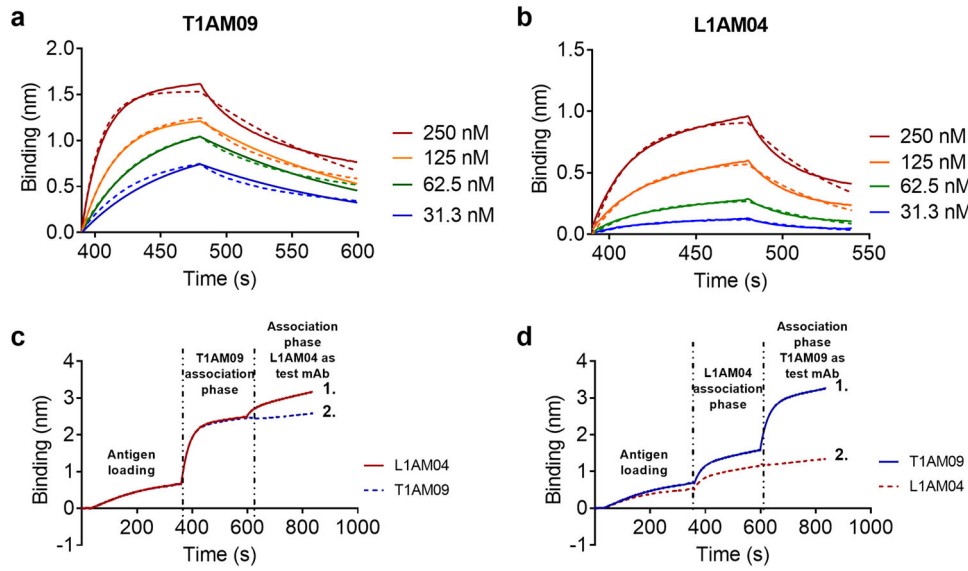

**Fig. 4 Bio-Layer Interferometry (BLI) and epitope binning quantitate affinity of mAbs to AM.** Binding curve of **a** T1AM09, and **b** L1AM04, generated by BLI - solid lines represent experimental data and dashed lines are statistically fitted curves. Two-phase binding experiment assessing **c** T1AM09 and **d** L1AM04 competition with each other. Solid line 1 shows the association curve with test mAb. Dashed line 2 is the association curve with self-control. .

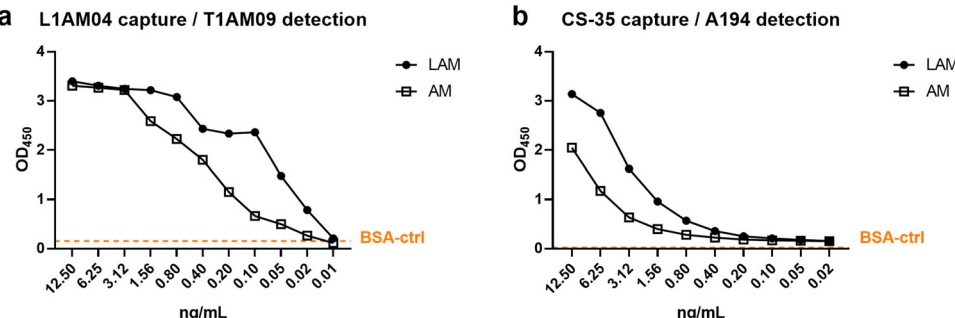

**Fig. 5 Combination of high-affinity human mAbs to distinct AM epitopes can detect low levels of LAM and AM in urine by sandwich ELISA.** Detection of serial dilutions of LAM and AM generated from the clinical *Mtb* strain CDC1551 and spiked into urine of a healthy volunteer by **a** L1AM04 (10 μg/mL) as a capture and T1AM09 (250 ng/mL) as a detection mAb, and **b** murine mAb CS-35 (10 μg/mL) as a capture and human mAb A194 (250 ng/mL) as a reference. The data shown are representative from two separate experiments.

**Table 2 Capture and detection of LAM from the urine of subjects with pulmonary TB by Sandwich ELISA.**

| Diagnosis[a] | Microbiology | | U-LAM[b] ELISA OD$_{450}$ | | U-LAM Detection[c] | GC-MS (ng/mL)[d] |
|---|---|---|---|---|---|---|
| | Smear[e] | Culture[f] | CS35/A194 | L1AM04/T1AM09 | | |
| Pulmonary TB | pos | *Mtb* | 1.649 | 1.531 | Positive | 25 |
| Pulmonary TB | pos | *Mtb* | 0.894 | 0.627 | Positive | 14.3 |
| COPD, Pneumonia | neg | neg | 0.233 | 0.210 | Negative | Not detected |
| Pneumonia, TB unlikely | neg | neg | 0.196 | 0.221 | Negative | Not detected |

[a]All four patients were HIV uninfected.
[b]Capture ELISA for the detection of LAM in urine (U-LAM).
[c]U-LAM detection above cut-off based on background values for healthy control urine for each mAb pair by capture ELISA (OD450 0.259 for CS35/A194 mAb pair and 0.335 for L1AM04/T1AM09 mAb pair, respectively).
[d]Quantification of U-LAM by gas chromatography-mass spectrometry (GS-MS) D-Arabinose detection method.
[e]Sputum smear microscopy for acid fast bacilli (AFB).
[f]Mycobacterial culture of sputum.

arabinan from the group of ManLAM motifs, Man$_3$Ara$_4$ and its related branched (Man$_3$)$_2$Ara$_6$. We demonstrate that these two mAbs are non-competing and are valuable for TB diagnostic and mycobacterial research applications.

Single B cell sorting has greatly accelerated our ability to generate human pathogen-specific mAbs[19–21]. Compared to other methods, such as hybridoma technology or B cell culturing, which are time-intensive and may not produce the naturally occurring mAbs from circulating B cells, AM-probe sorting is a targeted approach that can lead to the creation of recombinant human mAbs within weeks. Single B cell sorting is typically performed with proteins as probes and PBMC from recently vaccinated or infected individuals who have high levels of plasmablasts and/or memory B cells[37], but is less common with

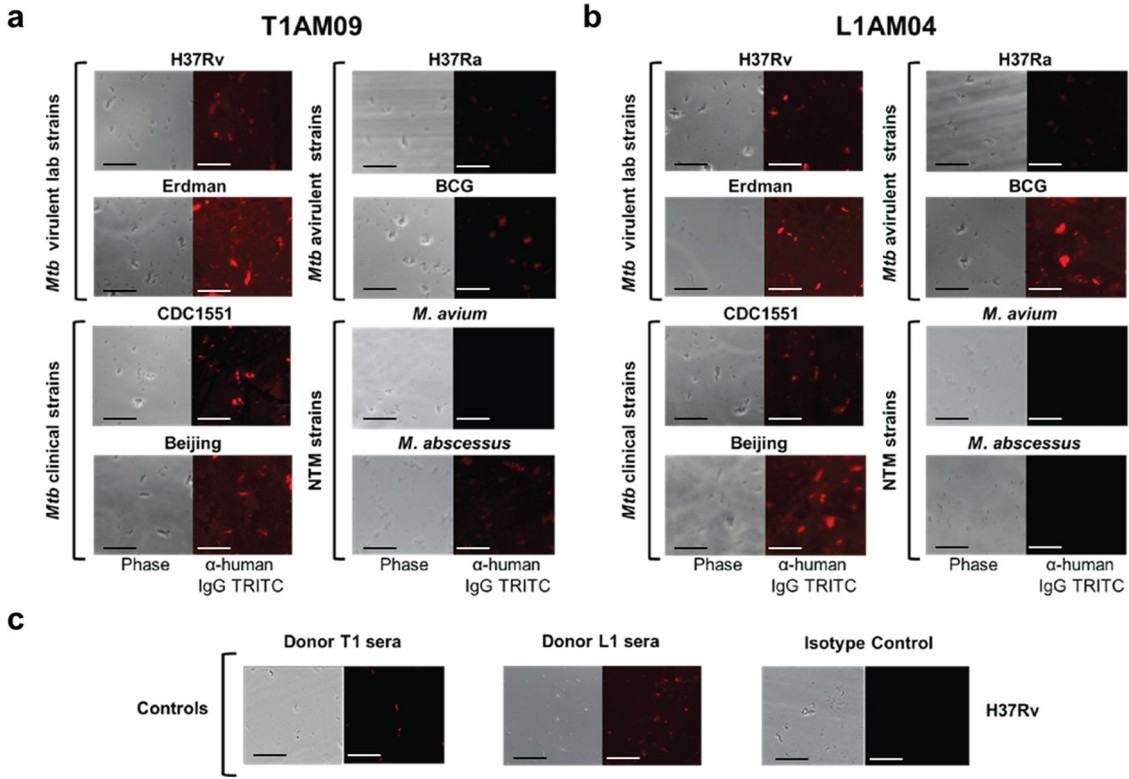

**Fig. 6 T1AM09 and L1AM04 show distinct binding to mycobacterial strains by immunofluorescence (mycobacteria were grown without detergent to preserve the capsule).** Binding of human IgG1 mAbs (10 μg/mL) and polyclonal serum IgG (1:100 dilution) to virulent laboratory (H37Rv and Erdman) and clinical strains (CDC1551 and Beijing) of *Mtb*, avirulent strains of the *Mtb* complex group (H37Ra and BCG Pasteur) and nontuberculous mycobacteria (*M. avium* and *M. abscessus*) for **a** T1AM09; **b** L1AM04; and **c** Binding of positive (sera from T1 and L1) and negative (isotype-matched mAb to a flavivirus) controls to H37Rv. Scale bar represents 10 μm.

glycans to which antibodies and B cell receptors have lower affinity (reviewed in Haji-Ghassemi et al.[16]). For TB, two recent studies used antigen-specific memory B cell sorting to generate mAbs to the *Mtb* proteins heparin-binding hemagglutinin (HBHA)[17] and phosphate-binding lipoprotein (PstS1)[38]. To date, only a few studies have successfully generated human anti-glycan mAbs using B cell sorting[23,24]. We show that anti-glycan mAbs, some with high affinity, can be generated using native *Mtb* AM as a probe. We believe that selecting donors with persistently high titers of anti-AM IgG over years was critical in increasing the probability of success.

The specific glycan epitope recognition of the two mAbs characterized here enhances our understanding of the breadth and diversity of human antibodies induced by *Mtb*. These specificities are distinct from anti-AM/LAM mAbs generated from TB patients via B cell culturing or from mostly *Mtb* and *M. leprae* immunized mice via hybridoma technology[26,35,39]. Because our mAbs were generated from memory B cells of asymtomatic individuals with LTBI or a history of TB exposure, they likely recognize clinically relevant *Mtb* surface glycan epitopes exposed during controlled *Mtb* infection in vivo. L1AM04 is a proof of concept that a fully human mAb can react with specific oligosaccharide epitopes sharing a trimannoside capped arabinan from the group of *Mtb* ManLAM motifs (reviewed in Turner and Torrelles[8]). ManLAM is present in *Mtb* and most other pathogenic mycobacterial strains and facilitates *Mtb* phagocytosis and intracellular trafficking, thereby contributing to *Mtb* survival within infected host cells (reviewed in Turner & Torrelles[8]). L1AM04's distinct reactivity with $Man_3Ara_4$ and the related branched $(Man_3)_2Ara_6$ but neither with single $(Man_1)$ nor the in *Mtb* most frequently occurring dimannose $(Man_2)$ capped arabinan motifs[40] suggests that these closely related ManLAM OS motifs are immunogenically distinct. Thus, L1AM04 will be particularly valuable for OS-specific investigations in future functional mAb studies. Moreover, this specific recognition of a trimannoside-capped arabinan allows for the distinction between *Mtb* and *M. avium* which produces predominately single mannose-capped arabinan compared to the dominant di-mannoside caps produced in pathogenic strains[41,42]. It further allows for the distinction between *Mtb* and *M. abscessus*, because capping motifs are absent at the nonreducing arabinan component of LAM from *M. abcessus*[43]. T1AM09 reacts with predominantly terminal arabinan motifs and, like L1AM04, is quite specific. These reactivities differ from the wider range of AM/LAM epitopes recognized by human IgG mAbs, such as A194 and P83A8, generated via B cell culturing from TB patients during uncontrolled *Mtb* infection[39]. The distinct glycan epitope speci-ficities of our two mAbs raise several questions: Could the speci-ficity of L1AM04 inform us about differences in AM epitopes exposed during latent compared to active *Mtb* infection? Given the changes in the amount of ManLAM during *Mtb* growth in vitro[44], does the shedding of LAM during active infection and its presence in *Mtb* vesicles[8,45–47] induce antibody clones more broadly reactive with AM epitopes? How are these AM/LAM components exposed on the *Mtb* surface and in *Mtb* vesicles during different states of *Mtb* infection in vivo? And, what other lessons can be learned from the differences in epitope recognition between L1AM04, T1AM09, and other anti-AM/LAM mAbs?

Conserved germline residues can be essential for glycan recog-nition by mAbs[33,48–51]. Some of the highest affinity anti-AM/LAM mAbs, including T1AM09 and L1AM04, are closely related to their germline and mutated about 5–10% at the nucleotide level[26,39]. VH3 germline restriction has been reported in mAbs to capsular

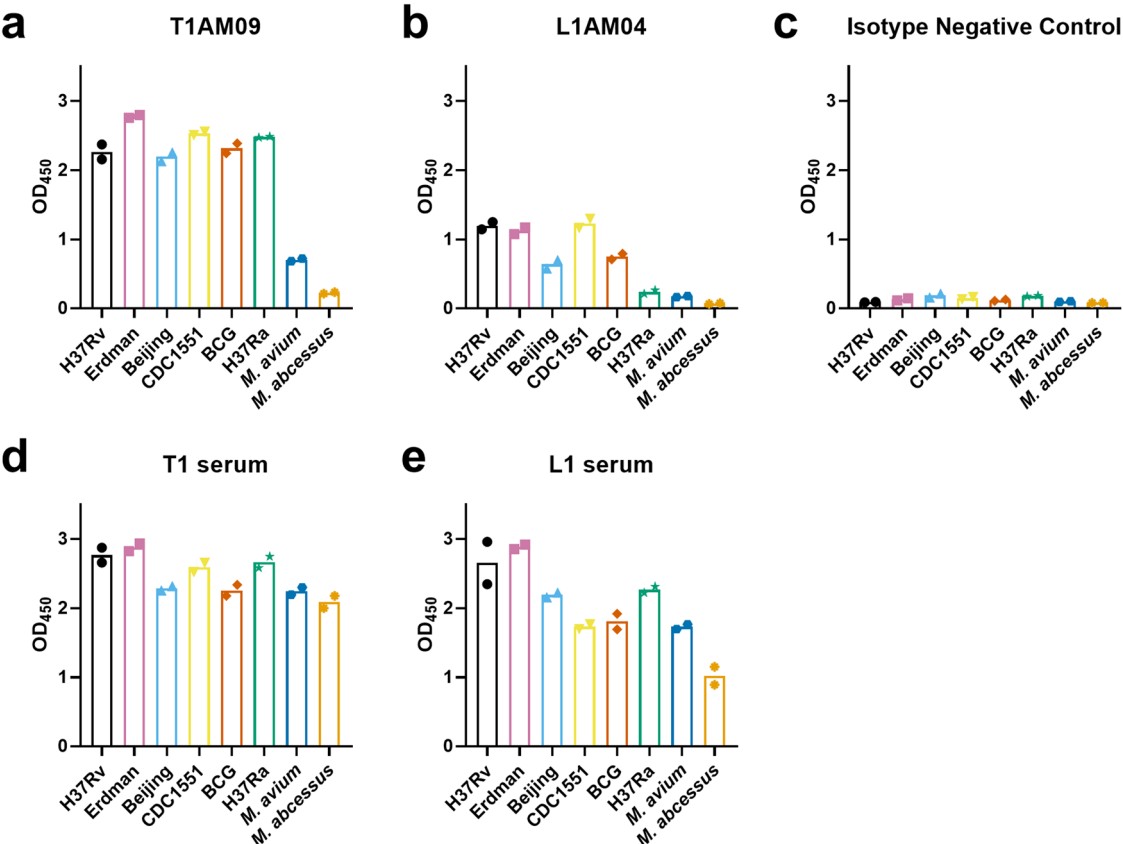

**Fig. 7 T1AM09 and L1AM04 show distinct binding to mycobacterial strains by whole-cell ELISA (mycobacteria were grown without detergent to preserve the capsule).** Binding of human IgG1 mAbs (50 µg/mL) and polyclonal serum IgG (1:20 dilution) to virulent laboratory (H37Rv and Erdman) and clinical strains (CDC1551 and Beijing) of *Mtb*, avirulent strains of the *Mtb* complex group (H37Ra and BCG Pasteur) and nontuberculous mycobacteria (*M. avium* and *M. abscessus*) assessed at optical density (OD) of 450 nm for **a** T1AM09; **b** L1AM04; **c** Isotype-matched negative control IgG1 mAb to a flavivirus; and positive controls, **d** Subject T1 serum; and **e** Subject L1 serum. The data shown are representative from two separate experiments.

polysachharides from *Cryptococcus neoformans*, *Haemophilus influenzae* type b (Hib), and *Streptococcus pneumoniae* (reviewed in Pirofski[52]). In humans, the VH3 germline accounts for about 35–40% of the antibody repertoire[53]. Greater than 50% of our human mAbs to *Mtb* AM generated in this study utilized VH3, suggesting that this germline might be over-represented among antibodies to AM/LAM, although, except for L1AM04, these VH3 mAbs had affinities too low to be further characterized. Our highest affinity mAb, T1AM09, is from the VH1-2 germline. VH1-2 germline antibodies are reported with broad reactivity to lipo-polysaccharides from several species of gram negative bacteria[54]. This germline is also associated with the class of broadly neutralizing mAbs to the HIV-1 CD4-binding site. While these mAbs do not all make direct contacts with glycans, they are reported to develop flexible complementarity-determining regions (CDRs) to tolerate glycan shields on these viruses (reviewed in Kwong et al.[55]). Recent studies have reported several over-represented heavy and light chain germlines and low mutation rates in antibody repertiores from primary malaria infection and COVID-19[49,50]. To the best of our knowledge, this is the first observation of distinct germlines and low mutation frequencies in high-affinity antibodies found in asymptomatic subjects exposed to *Mtb*. To which extent conserved germline residues and recurrent antibody sequences will play an important role in the protection against TB warrants further investigation.

High-affinity anti-AM/LAM mAbs can be valuable for TB diagnostic and research applications[34–36,56,57]. The current diagnostic modalities for TB and NTM include molecular tests, culture-based methods, and sputum smear microscopy[58]. None of these tests are suitable for use by community health workers—an urgently needed feature for resource-constrained settings where most TB cases occur[59,60]. A simple non-sputum-based point-of-care test for TB diagnosis is the U-LAM lateral flow Alere Determine TB-LAM Ag test. However, due to its poor sensitivity (it is only recommended for use in HIV–infected patients with CD4 cell counts under 100 cells/µl (reviewed in Shah et al.[3]). To develop new tests with improved sensitivity while maintaining high specificity, projects generating and assessing high-affinity anti-LAM mAbs for their U-LAM detection capacity are ongoing. Improved detection of U-LAM compared to the Alere test has been demonstrated in sandwich ELISA using the mAb pair CS-35 and A194[34,36], and in the FujiLAM lateral flow assay using the mAbs S4-20 and A194[35]. In a recent retrospective multicenter diagnostic accuracy study the FujiLAM lateral flow assay test accurately detected U-LAM in five times more HIV uninfected TB patients than the Alere test[35,36]. However, the requirement of a silver-amplification step, as needed for the FujiLAM assay, could reduce the point-of-care value by lowering the stability and shelf life of the assay, and increasing assay time and cost[61]. Furthermore, while mAbs can work well in a urinary LAM capture ELISA, they can have problems with aggregation and precipitation over time and/or conjugation (e.g. biotin), limiting their use for simple lateral flow formats needed the most for use by community health care workers in resource-constained settings. In addition, several of the mAbs recently investigated (e.g., CS-35 and A194) have broad epitope specificity leading to

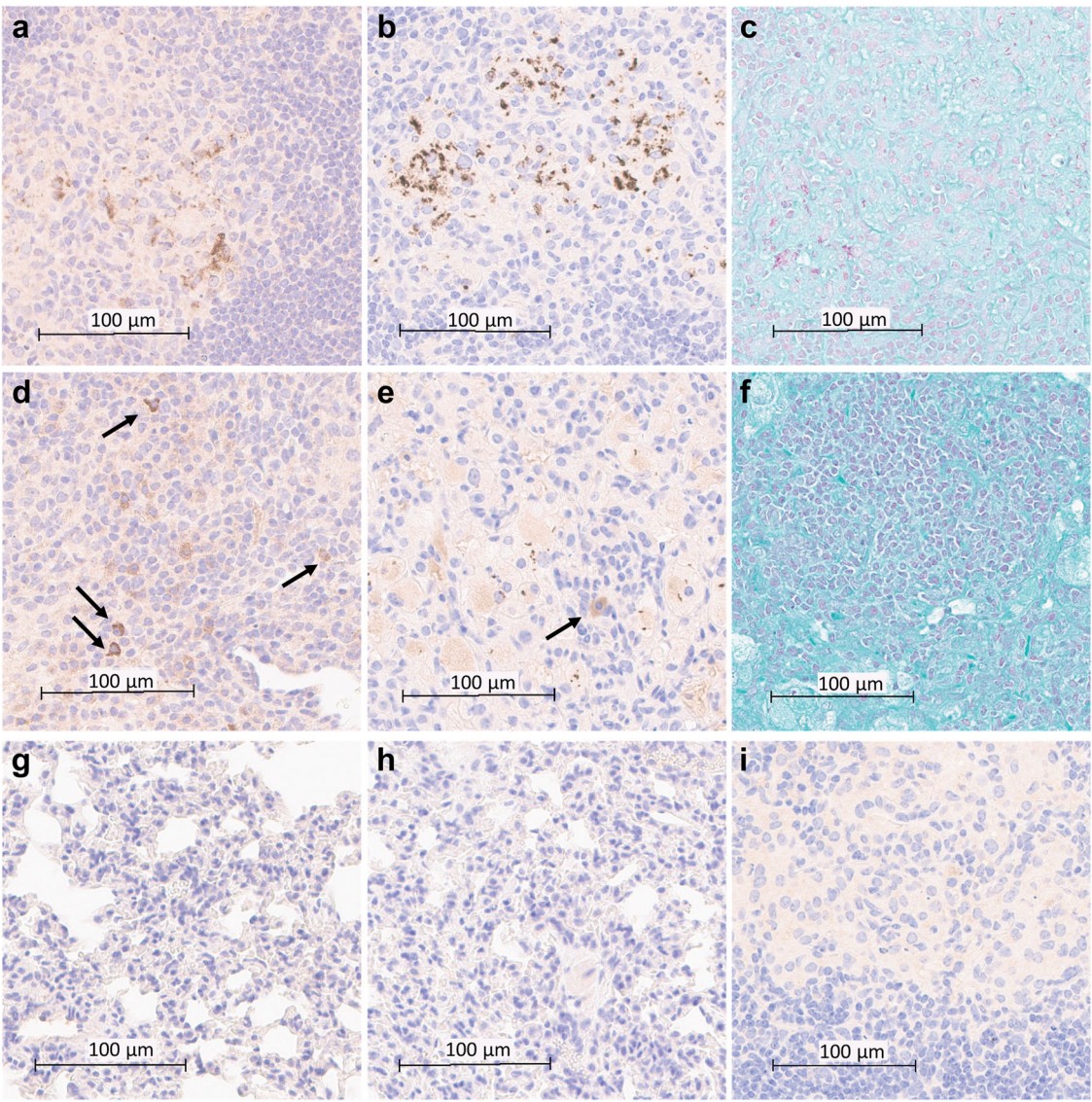

**Fig. 8 T1AM09 and L1AM04 detect extra- and intracellular *Mtb* and LAM in lung tissues of *Mtb*-infected mice.** Histology and immunohistochemistry of *Mtb* (CDC1551) infected murine lung (scale bar 100 μm) showing intra- and extracellular staining of *Mtb* by **a** T1AM09; **b** L1AM04; and **c** staining for Acid-Fast Bacilli (AFB); **d** Staining of intracellular LAM in *Mtb* (CDC1551) infected lungs (arrows indicate LAM within macrophages) by T1AM09; **e** Staining of intracellular LAM and single bacilli in *Mtb* Erdman infected lungs (arrow indicates LAM within macrophages) by L1AM04; and **f** lack of positive AFB staining of *Mtb* (Erdman) in approximately the same *Mtb* infected lung section as shown in the other figure panels. Overall lack of staining of non-infected murine tissue (scale bar 100 μm) by **g** T1AM09; and **h** L1AM04. **i** Lack of staining of *Mtb* (CDC1551) infected lung tissue by isotype-matched control mAb to a flavivirus (scale bar 100 μm). All mAbs were tested at 2 μg/mL.

cross-reactivity with NTMs[57,62]. Therefore, there remains an ongoing need for a spectrum of high-affinity anti-AM/LAM mAbs that do not compete with each other, react with different AM/LAM epitopes, and, ideally, are also *Mtb*-specific[56]. Because L1AM04 and T1AM09 are non-competing, work well as a binding pair for the detection of U-LAM, and have the potential to be combined with other high-affinity anti-LAM mAbs in U-LAM detection assays, they are a valuable addition to the existing library of anti-LAM mAbs to be further assessed for the improved detection of U-LAM in TB patients.

Detemining levels of U-LAM could also aid in the diagnosis of pulmonary or other NTM infections. There is an increasing emergence of NTMs that cause grave and difficult to eradicate infections in patients with underlying lung diseases (reviewed in Griffith et al.[63], Furukawa and Flum[64] and Chin et al.[65]). The two most common strains of NTMs that cause pulmonary infections

are *M. abcessus* and *M. avium*. In addition, patients with advanced HIV-infection are at risk for disseminated infection with *M. avium*[63]. Their differentiation from *Mtb*, diagnosis, and suceptibility testing can be time-consuming, and their treatment, consisting of multiple drugs for typically more than a year, varies from that of TB (reviewed in Griffith et al.[63], Furukawa and Flume[64] and Chin et al.[65]). Tools distinguishing NTM from TB, and determining mycobacterial burden, could not only aid in treatment decisions but could also serve as correlates of treatment success. The specificity of L1AM04 for virulent *Mtb* strains with little cross reactivity with *M. abcessus* or *M. avium*, combined with the reduced binding of T1AM09 to *M. abcessus* and *M. avium* by IF and ELISA, respectively, could be valuable in distinguishing between *Mtb* and these two strains in pulmonary and other NTM infections, highlighting the role of glycan epitope specificity in the binding of mAbs to NTMs.

The AFB staining using the Ziehl–Neelsen or Kinyoun stain is most commonly used to detect mycobacteria in tissues and body fluids, such as sputum[66,67]. However, AFB staining is neither specific for *Mtb* nor sensitive—typically over 100 mycobacteria per mililiter of tissue or body fluid are needed to visualize bacilli by bright-field microscopy[66]. Moreover, during natural infections in humans and experimental infections in mice, non-replicating *Mtb* bacilli stain acid-fast negative[68]. We show that IHC using T1AM09 or L1AM04 could be an alternative approach to improve sensitivity and specificity of both intra- and extracellular *Mtb* detection compared to AFB staining. We further show that T1AM09 and L1AM04 can detect intra- and extracellular LAM. LAM is shed by *Mtb* during infection[8,45–47], and its detection could inform further immunologic studies. Therefore, both mAbs could provide valuable tools for histology and research applications.

The focus of our studies was limited to the two high-affinity anti-AM/LAM mAbs T1AM09 and L1AM04. Because glycans are less hydrophobic than some protein side chains, which drive protein–protein recognition, glycan-targeting antibodies tend to have lower affinities than protein-targeting antibodies[16,69]. Indeed, the majority of the mAbs we isolated in this study exhibited weak reactivity, limiting their characterization. BLI measurements indicate that T1AM09 and L1AM04 bind with $K_{D,app}$ values in the mid-nanomolar range with strong binding also seen by various other methods such as ELISA, microarray, and fluorescence microscopy. Developing a sorting strategy using oligosachharide motifs is feasible, and might offer a more focused method leading to further specific mAbs[23,24], but whether such strategy would yield high-affinity mAbs remains to be explored. Oligosacharide sorting was beyond the scope of the present studies because the synthetic generation of sufficient quantities of AM oligosacharides is time-consuming and the biophysical considerations for the binding of B cells to oligosaccharide probes (e.g., the importance of mutlivalency) have yet to be better understood. Further assessment of mAbs CDR flexibility and a molecular-level undertstanding of how the AM epitopes are bound, is also needed, but was beyond the scope of the current studies. Understanding the paratope of our anti-AM/ LAM mAbs could contribute to the discovery of important mutations and motifs that may improve the binding of some of the existing mAbs.

In conclusion, using the *Mtb* glycan AM as a probe and high anti-AM IgG titer asymptomatic individuals infected with or exposed to *Mtb*, yields both high affinity AM glycan epitope cross-reactive and uniquely specific mAbs with value for TB research and diagnostics. The distinct epitope specificity of these compared to other anti-AM/LAM mAbs enhances our understanding of the spectrum of antibodies elicited to AM/ LAM in humans, and is important for future functional studies against *Mtb*.

## Methods

**Subjects and samples**. Sera and PBMC were collected from subjects enrolled in a TB study (#2006-428) approved by the institutional review board of the Albert Einstein College of Medicine/Montefiore Medical Center. Study subjects gave written informed consent prior to their enrollment. Sera and PBMCs were processed within a few hours of collection and frozen until use at −80 °C.

For spiked urine, anonymized urine control samples were obtained from healthy volunteers from a TB non-endemic region in Colorado State University (CSU), aliquoted and frozen at −80 °C until further use (approved by CSU IRB under approval numbers 15–104B and 09-006B). All urine specimens were sedimented by centrifugation and the supernatants were stored at −80 °C within a few hours of collection.

For urine obtained from patients, anonymized archived urine samples used in this study were provided by the Foundation for Innovative New Diagnostic (FIND, Geneva) and stored at CSU. The urine samples were collected from patients presenting to clinics in Vietnam with respiratory symptoms consistent with

pulmonary TB and prior to the initiation of treatment[57]. Final diagnosis (TB vs. non-TB) was established by microscopy, at least two sputum cultures and clinical and radiologic examinations. TB was defined based on a positive sputum culture for *Mtb* and non-TB was defined as being smear and culture negative on all samples and having improved clinically/radiologically without TB-specific therapy. For our study, testing included urine samples obtained from four patients (two with TB and two with non-TB)[70]. All four patients were HIV uninfected, two were diagnosed with sputum AFB smear-positive and culture-confirmed pulmonary TB. The other two patients had negative sputum smears and mycobacterial cultures, and were diagnosed with other respiratory diseases than TB, one with pneumonia, the other with pneumonia and chronic obstructive lung disease.

**Culturing of mycobacteria and generation of capsular AM**. Various strains of the *Mtb* complex group comprised of laboratory strains (H37Rv, H37Ra), clinical strains (CDC1551, Beijing) and the TB vaccine strain *M. bovis* Bacillus Calmette–Guérin (BCG) were pre-cultured in Middlebrook 7H9 broth supplemented with 0.05% (v/v) tyloxapol and 10% (v/v) oleic albumin dextrose catalase enrichment to reach stationary growth phase ($OD_{600}$ of 0.5–1.0). Similarly, non-tuberculous mycobacterial strains (*M. abscesses* and *M. avium*) were cultured. To allow for mycobacterial capsule formation, the pre-culture strains were inoculated in minimal medium without detergent at 37 °C for 3 weeks[14,71]. Minimal medium consisted of 1 g/l $KH_2PO_4$, 2.5 g/l $Na_2HPO_4$, 0.5 g/l asparagine, 50 mg/l ferric ammonium citrate, 0.5 g/l $MgSO_4 \times 7\ H_2O$, 0.5 mg/l $CaCl_2$, 0.1 mg/l $ZnSO_4$, with or without 0.05% tyloxapol (v/v), containing 0.1% (v/v) glycerol, pH 7.0[71]. Capsular polysaccharides were isolated by physical disruption of cells using glass beads, followed by clarification, and lyophilization[14]. AM was separated from the other capsular components using chloroform:methanol:water extraction (1:1:0.9) and isolated and purified[13,14]. Collected fractions were assayed for carbohydrate content by the phenol-sulfuric acid assay. AM-positive fractions were tested using a standard ELISA and positive control antibody (murine mAb CS-35, BEI NR-13811). The concentrations of AM and LAM were confirmed by GC-MS calculating the ratio of arabinose to mannose[34].

**Isolation of AM-specific B cells**. To identify AM-positive B cells by standard B cell immunophenotyping and fluorescence-activated cell sorting (FACS)[19,20], AM was activated by 1-cyano-4-dimethylaminopyridinium tetrafluoroborate (CDAP) and conjugated to biotin (1:20). The AM-probe was then used with the antibody cocktail for cell staining (Fig. S2)[14,72]. Lymphocytes were first identified, then memory B cells were further enriched based on CD20 + CD27+ for subject T1. Secondary staining using streptavidin labeled with phycoerythrin (PE) was used to detect the AM-positive population. For subject L1, to increase the yield and quality of single B cells (AM + IgG+), we incorporated a dump channel consisting of live/ dead stain (Zombie Aqua, Biolegend), CD3, and CD14 to exclude dead cells, T cells, and monocytes[73]. Subsequently, CD19 was incorporated into the antibody cocktail to ensure consistency with similar sorting strategies of mAbs to oligo-saccharides from other pathogens[23,24]. This second strategy was also subsequently repeated for T1. Fluorescence-minus-one (FMO) controls were used to set the sorting gates for double positive (AM+, IgG+) B cells (Supplementary Fig. 3) which were sorted into a 96-well plate containing RNAse-inhibiting lysis buffer[20].

**Generation of mAbs**. Reverse transcription PCR (RT-PCR) of lysed single B cells was used to synthesize complementary DNA (cDNA). Immunoglobulin genes for the heavy and light chains of the variable (V) region (Ig $V_H$ and $V\kappa$) were amplified using random hexamer primers and then sequenced. Nested PCR was used to clone Ig genes into IgG1 heavy- and light-chain expression vectors which were co-expressed by transfection of 293 HEK cells[19,20]. Germline alleles were determined using the IMGT database (http://imgt.org).

**Antibody binding assays**. We used ELISA to determine the binding of mAbs to capsular AM, cell wall LAM, LM, and *Mtb* fractions. Polysaccharide/glycolipid antigens (AM, LAM, LM) were coated at 10 μg/mL and *Mtb* fractions (cell wall, MEM, and culture filtrate) were coated at 4 μg/mL. Plates were blocked with 3% BSA in PBS for 1 h. After blocking, the plates were washed with TBST (200 μL × 5) and were incubated for 1.5 h at 37 °C with serially diluted mAbs. Following a second wash, 50 μL of anti-human IgG Fc–Horseradish Peroxidase (HRP; 1:4000) was incubated for 1 h at 37 °C. After the final wash, 50 μL Ultra TMB-ELISA chromogenic substrate (Fisher Scientific) was added to the plates and incubated for 5 min. The reaction was stopped by the addition of sulfuric acid (Sigma) and the OD was read at 450 nm.

To visualize the binding of mAbs to fixed *Mtb* with an intact capsule, we used immunofluorescence microscopy. Briefly, mycobacteria from the *Mtb* complex group and the NTMs *M. abscessus* and *M. avium*, cultured in conditions to preserve the capsule[15], were fixed with 2% PFA and coated onto Poly-L-lysine slides. MAbs, incubated at 10 μg/mL, were labeled with tetramethylrhodamine (TRITC)-labeled goat-anti human IgG and viewed with a Zeiss observer microscope[13]. Donor sera (subject T1 and L1) were used as positive controls and human IgG1 mAb to a flavivirus was the matched isotype negative control.

We quantified the binding of the mAbs to different mycobacterial strains by a whole-cell ELISA as described[74,75]. A human IgG1 mAb to a flavivirus was used as

the matched isotype negative control and donor sera (subject T1 and L1) were used as positive controls. Briefly, *Mtb*, cultured in conditions to preserve the capsule[15], were concentrated by centrifugation 3000 rpm for 20 min at 25 °C and washed 1× with PBS. For each mycobacteria sample, independent bacterial aliquots were used for protein determination and coating a 96-well plate (Immunolon). Prior to use in ELISA, volumes of suspensions were adjusted based on their protein concentration described[74,75]. All plates were coated with 20 µg/ml mycobacteria in PBS overnight at 4 °C and then blocked with BLOTTO (Thermo Fisher) for 1 h at 37 °C. After blocking, the plates were washed 5× with TBST and incubated for 1.5 h at 37 °C with mAbs at 50 µg/mL or sera at a 1:20 dilution. Following a second wash, anti-human IgG Fc–Horseradish Peroxidase (HRP; 1:4000) was incubated for 1 h at 37 °C. After the final wash, 50 µL Ultra TMB-ELISA chromogenic substrate (Fisher Scientific) was added to the plates and incubated for 30 min. The reaction was stopped by the addition of sulfuric acid (Sigma) and the OD was read at 450 nm.

**Bio layer interferometry (BLI).** Initially, we tested the binding of the anti-AM mAbs to the biotinylated probe with the OctetRed$^{TM}$ system (ForteBio, Pall LLC) as described[22]. Briefly, biotinylated AM was loaded on to streptavidin biosensor tips at 0.5 µg/mL. The loaded antigen was dipped into serially diluted mAb. Binding curves were generated and used to calculate dissociation constants. To estimate values for the $k_{on}$ (association rate constant), $k_{off}$ (dissociation rate constant), and $K_{D,app}$ (apparent equilibrium dissociation constant), we used a global data fitting 2:1 binding model. The 2:1 heterogenous ligand model assumes analyte binding at two independent ligand sites. Each ligand site binds the analyte independently and with a different rate constant (Octet ref Application Note 14). For epitope binning experiments, a double-phase binding set-up was used; biotinylated-AM was first immobilized on a streptavidin sensor and then allowed to equilibrate in a solution containing the first mAb. After the binding plateaus, the biosensor is dipped into another well containing equimolar amounts of competitor mAb or several controls. The observation of a second association curve suggests that there are available binding sites on AM to the competitor mAb, but not the mAb itself. Data were analyzed using ForteBio Data Analysis Software 9.

**Glycan microarrays.** To determine the mAb reactivity to AM glycan epitopes, we used a recently developed glycan microarray comprised of 63 synthetically generated oligosaccharide fragments[13,15,30]. Briefly, microarray slides were blocked with 3% BSA (GEMINi Bio-Products, standard grade) in PBS at 4 °C overnight, followed by mAbs incubated at 5 µg/mL for 4 h at 37 °C. After washing (6×) with 0.1% PBS-T, the slides were first incubated with goat anti-human biotin-labeled IgG or goat anti-mouse biotin-labeled IgG (Southern Biotech, AL) 1:1000 in 1% BSA in PBS and followed by a streptavidin probe tagged with SureLight®P3 (Cayman Chemicals, MI) 1:200 in 1% BSA in PBS at 37 °C for 2 h. The GenePix 4000 Microarray scanner system (Molecular Devices, CA) was used for scanning. We analyzed images using GenePix Pro 7.3.0.0 to measure median pixel intensity (MPI) and neighboring background pixel intensity (BPI) of individual spots. The median fluorescence intensity (MFI), representing AM-epitope specific mAb reactivity, was calculated using the MPI minus the BPI and averaged from the triplicate spots[13,15].

**Sandwich LAM ELISA.** The mAbs T1AM09 and L1AM04 were tested for their use as capture and/or detection antibodies in a capture sandwich ELISA[34]. The pair of anti-LAM mAbs CS-35 (murine) and A194 (human) were used as a positive control to access capture and detection of serially diluted LAM in urine obtained from a healthy volunteer from a non TB endemic region and used to generate a standard curve. Capture antibodies (CS-35 and L1AM04) were used at 10 µg/mL in PBS and coated on a 96-well high binding polystyrene plate and incubated at 4 °C overnight. Urine from a healthy volunteer was spiked with known amounts of *Mtb* CDC1551 LAM (quantified by GC-MS), briefly incubated (30 min) at room temperature (RT) for the complexation of LAM and urinary proteins and then stored overnight at −20 °C[34,70,76]. Testing also included clinical urine samples obtained from four patients who presented with respiratory symptoms to an ambulatory health clinic in Vietnam. Un-spiked urine was used as a background negative control and for the determination of cut-off values for positive LAM detection in clinical samples for each mAb pair. Clinical samples were pretreated with Proteinase K and the supernatant used for ELISA[34,76]. After overnight incubation, the antibody-coated plate/s and the LAM spiked urine samples along with the clinical samples were brought to RT and the antibody-coated plate/s blocked for 1 h at RT. After blocking, the plates were washed briefly (200 µL × 2), samples were added to the appropriate wells and incubated at RT for 90 min. The plates were washed (200 µl × 10) followed by incubating for 90 min at RT with the biotinylated detection antibodies (A194 or T1AM09) at a final concentration of 250 ng/mL. Following a second wash, 100 µL of 1:200 dilution of Streptavidin–Horseradish Peroxidase (HRP; R & D Systems) was added to the plates and incubated as per protocol. After the final wash, 100 µL Ultra TMB-ELISA chromogenic substrate (ThermoFisher Scientific) was added to the plates and incubated for 30 min. The reaction was stopped by the addition of sulfuric acid (Fisher Scientific) and the OD was read at 450 nm. The samples were run in duplicates and plotted against the standard curve to obtain the LAM detection status based on the no LAM cut-off values.

**Immunohistochemistry staining.** Tissue specimens were fixed in 10% buffered formalin for 72 h in 10× the volume of tissue. After fixation, paraffin-embedded blocks were prepared, and immunostaining was performed. Staining was conducted on a Leica Bond RXm automated staining platform. For antigen retrieval, slides were heated in a pH9 EDTA-based buffer for 25 min at 94 °C, followed by a 30 min RT antibody incubation (2 µg/mL, humanized clone T1AM09 or L1AM04). Antibody binding was detected using a rabbit anti-human secondary antibody pre-adsorbed against mouse (Southern Biotech 6145-01) and an anti-rabbit HRP-conjugated secondary polymer, followed by chromogenic visualization with diaminobenzidine (DAB). A Hematoxylin counterstain was used to visualize nuclei. AFB staining was used as a positive control and performed as described[77]. Stained slides were viewed using Aperio ImageScope (v12.4.3.5008).

**Statistics and reproducibility.** Statistical analysis was performed using Prism software version 7.04 (GraphPad). For overall group comparisons, a one-way ANOVA or the nonparametric Kruskal–Wallis test was used depending on the distribution of the data. A $p$-value <0.05 was considered statistically significant. All experiments contained duplicates and were repeated at least two times.

**Reporting summary.** Further information on research design is available in the Nature Research Reporting Summary linked to this article.

## Data availability

The accession numbers for the nucleotide and protein sequences for variable regions of T1AM09 and L1AM04 are GenBank: MZ600149 - MZ600152. Glycan array data were deposited in the NCBI Gene Expression Omnibus (GEO) database with accession number GSE180517. Source data all of the figures were deposited at https://doi.org/10.5061/dryad.dv41ns200.

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

## Acknowledgements

This work was supported in part by funds from the National Institute of Health (NIH)/ National Institute of Allergy and Infectious Diseases (NIAID) to J.M.A. (AI146329, AI127173, and AI117927), J.R.L. (R01-AI125462), D.C., and A.G.A. (R01 AI132680). E.I. was supported by NIH Institutional Clinical and Translational Science Award (U54) grant 5TL1TR001072-06 and NIH T32 Fellowship in Geographic Medicine and Emerging Infectious Diseases (2T32AI070117-14). R.J.M. was supported by NIH Medical Scientist Training Grant T32-GM007288 and a training fellowship F30-AI150055. T.L.L. is thankful for the support from the Canadian Glycomics Network (SD-1). We gratefully acknowledge technical assistance from the Einstein Flow Cytometry Core (P30CA013330).

## Author contributions

E.I. and J.M.A. conceived and designed the study, analyzed and interpreted all data, performed literature review, and wrote the manuscript. D.C., T.L.L., and J.R.L. contributed to study design, data analyses and interpretation, and reviewed and edited the manuscript. E.I., D.T.C., and T.C. performed B cell sorting and antibody binding assays. R.J.M. and D.H., performed B cell sorting and kinetic experiments, A.G.A. performed the urinary LAM sandwich ELISAs. M.J. synthesized the oligosaccharide motifs for the glycan array. D.C., T.L.L., J.R.L., and J.M.A. provided resources. All authors reviewed and approved of the manuscript.

## Competing interests

The authors declare the following competing interests: The mAbs described herein are the subject of US patent applications entitled "High-Affinity Mycobacterium Tuberculosis Capsule-Specific Human Monoclonal Antibody" (PCT Application No. 17/047,256, US Provisional Application No. 63/199,235 & 17/247,532) with E.I., D.H., T.C., J.R.L., and J.M.A. as co-inventors. Data in this paper are from a thesis (E.I.) to be submitted in partial fulfillment of the requirements for the Degree of Doctor of Philosophy in the Biomedical Sciences, Albert Einstein College of Medicine. The remaining authors declare no competing interests.
