## [Peer Review File · Communications Biology]

Human monoclonal antibodies to M. tuberculosis arabinomannan recognize distinct glycan epitopes - implications for TB diagnostics and researchReviewers' comments:

Reviewer #1 (Remarks to the Author):

Human monoclonal antibodies to *M. tuberculosis* arabinomannan recognize distinct glycan epitopes - implications for TB diagnostics and research

Overall evaluation:

This is a well written manuscript that is based on previous work by the same group where the authors found that asymptomatic *Mtb*-LTBI donors have anti-AM serum antibodies that inhibit the growth of *Mtb*. Based on this previous finding, in the current study the authors isolated the monoclonal antibodies responsible for the anti-mycobacterial activity, from two donors; T1 – the donor that exhibited the strongest anti-AM antibody titers, and donor L1, an IGRA negative donor that exhibited weaker antibody anti-AM titers.

While this is the first study to clone anti-AM monoclonal antibodies from LTBI humans using single cell sorting technology, mAbs against *Mtb* glycolipids LAM and AM were described many times before, including human mAbs. One mAb is used as a positive control in this study and has several orders of magnitude better binding. Moreover, as opposed to the serum of the donors that exhibited interesting anti-mycobacterial activity, none of the antibodies cloned during this study had inhibiting anti-MTB functions, why is this unclear to me. Overall, though cloned from human origin, the 2 antibodies isolated in this study are not directed against significantly new targets, are not *Mtb*-inhibiting, and have a weak binding compared to previously described mAbs. The authors speculate, that due to their LTBI-human origin, and different glycan signature binding, these mAbs can be more useful in discriminating LTBI from Active infections, however this is not demonstrated in this study.

Detailed evaluation:

The authors start with describing their staining, gating and sorting strategy. Can the authors comment why the gating was done using two different strategies? While for T1 the authors gated on CD20/CD7-pos, AM-pos, IgG-pos, for donor L1 the authors gated on CD19-pos/CD3-neg, AM-pos, IgG-pos. If two different strategies were used, a comparison between the two samples cannot be carried out. There is also a clear compensation problem visible on the last T1 gate – the one that was sorted... Figure 1a,right. This might explain why the approximately 80% of the cells that were sorted did not contain functional antibodies when produced. Also, it looks that only 2-3 mAbs produced from each patient were actually specific for the probe. This clearly indicates that the entire procedure was not efficient and there was a lot of noise during the staining and sorting steps.

Based on ELISAs and BLI measurements the antibodies T1AM09 and L1AM04 have weak binding, at least compared to previously described anti-AM mAb CS-35. Nevertheless, the authors repeatedly write that the new mAbs have "high affinity". Except the BLI, and glycan fingerprints, where previously isolated anti-AM mAb CS-35 was included for comparison, and exhibited 1000-10000-fold better binding, and broader glycan recognition, direct competition or comparison experiments between T1AM09 and L1AM04 and other anti-AM mAbs (CS-35) were not done. Therefore, it is not clear whether the new mAbs bind stronger, or to new sites. Interestingly, in Figure 5 using both these mAbs for detection of uLAM was superior to the excising setup of CS-35/A194. However, this was not shown in *Mtb*-infected donor, but rather in a healthy volunteer where LAM was diluted. Also several negative controls are lacking (isotype control, or negative control antigen)

In Figure 6 The authors write that there is "strong binding" to various strains of *Mtb*, based on microscopy. However, the degree of staining signal is not clear from the figure. For example, the authors write that there is binding to *M. abscessus*, whereas the staining looks very weak. What % of the bacteria is bound by the mAbs? How is it compared to the anti-AM mAb CS-35. The same is true for Figure 7.

In the Discussion section the authors discuss germline restriction. They write that one of their findings is that Vh3 is overrepresented in anti-*Mtb* mAbs (similarly to COVID-19 and Malaria).

However, Vh3 is the most common Vh within the human Vh gene repertoire, therefore was this claim based on normalization to the individual frequency on each Vh within the Vh gene repertoire? This analysis should be included in the figures, or this claim cannot be made. Also, the claim that Vh1-2 antibodies in HIV-1 "are associated with broad polyclonal reactivity to lipopolysaccharides" is completely untrue. While some Vh1-2 mAbs maybe bind sugars on top of the heavily glycosylated HIV-1 (which are in a completely different configuration as the authors themselves note from the mycobacteria LAM and AM), the most potent anti-HIV-1 CD4 bindings site Vh1-2 arising neutralizing antibodies are NOT glycan dependent (VRC01, 3BNC117, 3BNC60 and more)...

Minor comments:

- Figure 2 – no negative or positive control antibody is shown. Also figure fonts should be the same size.
- In Supplementary FACS plots, axis labels should be clearly indicated.
- Glycan fingerprint from Supplementary Figure 3 should be included in main figure 3 for comparison
- Figure 6 and Supplementary figure 4 – there is no scale

Reviewer #2 (Remarks to the Author):

Arabinomanna and lipoarabinomannan are polysaccharide components of Mycobacterium capsule and cell wall. They play important roles in pathogenesis, and antibodies to these glycans can be useful for a variety of purposes. In this study, the authors use single B cell sorting to identify monoclonal antibodies to AM/LAM. They use a variety of techniques, including ELISA, BLI, glycan microarrays, cell binding, and IHC to evaluate the antibodies. Two antibodies of special interest could distinguish between virulent and non-tuberculosis strains, and they could act as a capture and detection pair for analyzing LAM in urine. Overall, the paper is well written, and the new antibodies are useful and fascinating. I recommend publication pending some minor revisions.

In the Discussion section, the authors state that VH3 germline genes are overrepresented for human mAbs to MtbAM. I think this statement might need to be clarified or revised. VH3 is one of the most common VH subgroups in humans - from one paper (Arnaout Plos one 2011), VH3 is about 35-40% of all human VH. So, a random sample of human antibodies might have roughly the same proportion of VH3 as their mAbs. Therefore, I'm not sure I would call them overrepresented, or perhaps they might clarify.

Figure 3. For the glycan structures, I would probably add a description of what the different bond direction mean. I suspect most non-glyco people will not understand that the direction means something. Maybe state in the legend that all bonds are alpha unless listed in the figure as beta.

As a reader, I would like the full sequences of all the antibodies to be listed in the Supp. What sort of amino acid mutations occur? where are they located? Also, it would enable others to evaluate and use the antibodies. Of course, there may be IP issues. I will leave it to the Editors to decide if the sequences are required or not.

Experimental details: In many places throughout the Methods section, the sources of chemicals/antibodies/proteins, working concentrations, and buffers are not specified. For example, several places list a dilution of 1:200 but the buffer is not listed. If BSA is used in the buffers, list the source, quality, and working concentration. As a second example, the buffer and concentration of the SA-Cy5 secondary for the glycan microarray assay are not listed. Please add this information throughout the Methods section.

Reviewer #3 (Remarks to the Author):

This well-written report describes the development of monoclonal antibodies from B cells with very high affinity for mycobacterial arabinomannan (AM) from two asymptomatic BCG-vaccinated, previously-infected or *M. tuberculosis* (Mtb)-exposed, skin-test positive individuals whose serum was previously shown to have high anti-arabinomannan IgG titers that when passively transferred to Mtb-infected mice reduced mycobacterial burdens. The immunoglobulin variable regions from AM-positive B cells from these two individuals were cloned into an IgG1 expression vector in HEK293 cells and a single mAb (T1AM09 or L1AM04) with the strongest AM binding by ELISA was selected from each group for further glycan-binding characterizations. The report's claim that these mAbs have exceptionally high affinity to Mtb AM and lipoarabinomannan (LAM) and are noncompeting is supported by glycan array analyses which identify unique motifs recognized by each mAb (Fig. 3) competition binding studies showing that binding of one to AM does not exclude binding of the other (Fig. 4), and combining both in a urinary-LAM sandwich ELISA test demonstrating greater sensitivity for both AM and LAM than a commercially-available U-LAM test consisting of CS-35 capture and A194 detection mAbs using urine spiked with LAM from Mtb strain CDC1551 (Fig. 5). Microscopic examination of binding by mAbs T1AM09 and L1AM04 to distinguish virulent Mtb strains from non-tuberculous species was less compelling (Fig. 6). Both mAbs can be used to detect Mtb in infected mouse lung tissue (Fig. 7).

Lines 40-42: While figure 6 does show strong detection of the virulent Mtb strains and BCG with either T1AM09 or L1AM04, weak detection of *M. abscessus* only by T1AM09, and neither causing H37Ra or *M. avium* to fluoresce red, it is unclear if the lack of or limited detection of the nontuberculous strains is due to reduced clumping of these strains, reduced production of the AM targets of these mAbs, or a microscope resolution or intensity setting issue. The case for detection could be made stronger by including some panels in which the strains also expressed GFP. AECOM has renowned Mtb investigators who have GFP--encoding plasmids for use in mycobacteria and may even have these species already engineered to express GFP.

Lines 261-2: lack of positive staining by AFB outside of the inflammatory region in panel? Panel F appears to have a large number of infiltrating cells. Please clarify what is meant by outside the inflammatory region.

Figure 7. Why are the uninfected mouse tissues (panels g-i) at a much lower resolution than the Mtb-infected tissues? For apples-to-apples comparison of the extent of background staining, similar resolution should be used. It would be especially interesting to know if the diffuse tan staining in the mAb is the result of the presence of the bacteria and how it compares to the isotype control.

Lines 78-81: Please clarify this sentence

Line 157: glycopeptidolipids?

Line 225: somewhere in the legend to figure 5 it should be stated that this is a sandwich ELISA.

Line 240: Indicate the magnification used in this figure.

Lines 302-304: Please clarify this sentence.

Line 350: For clinical relevance, the antibody pair must be able to detect AM or LAM epitopes that remain intact. While the results with the LAM-spiked urine look great in figure 5, it was disappointing that no comparisons of the L1AM04/T1AM09 pair to CS-35/A194 were performed with urine from TB-positive humans or from urine collected from Mtb-infected animals if human samples are not available.

Line 374: extraceulluar?

Line 412: which minimal medium was used to culture the strains?

Supplementary Figure 3: Please label to top of the panel A drawing as done in Figures 3 A and B.

Supplementary Figure 5: Is it possible to conclude that this staining method lack sensitivity based on the images in this figure alone? What is the positive control for Fite's method? Why is only the pink color with the arrow in panel B the only positive staining? What is the pink staining above it?

We thank the reviewers for their time, comments, and input. We have summarized and responded to main concerns upfront, followed by point-by-point comments and responses to each reviewer below. We have performed additional work and have revised our manuscript (changes in response to reviewers and for better clarity are highlighted).

Reviewers' main criticisms/concerns were:

- i) The value of generating and characterizing human mAbs to AM/LAM when there are already other anti-AM/LAM mAbs, including one murine mAb (CS35) with exceptionally high affinity (Reviewer 1).
- ii) Lack of demonstration that mAbs can distinguish between latent and active Mtb infection (Reviewer 1).
- iii) The lack of testing and comparing our mAbs for binding with samples from TB patients (Reviewers 1 and 3).
- iv) Quantification of binding of our mAbs to mycobacterial strains, especially the nontuberculous mycobacteria *M. abscessus* and *M. avium* (Reviewers 1 and 3).

Responses:

- i) *The mycobacterial surface antigens AM/LAM and antibodies against them are critically important in TB pathogenesis, pathogen-host interactions, and diagnostics. As polysaccharide and glycolipid antigens, AM and LAM exhibit a more fluid nature than protein antigens, thereby presenting the immune system with multiple orientations. As outlined throughout our manuscript, our understanding of the spectrum of antibody reactivity to AM/LAM glycan epitopes is limited, especially in humans who, in contrast to mice and other small animal models, can control Mtb in a latent infection. Both knowledge of antibody AM/LAM epitope specificity and the generation/characterization of mAbs to distinct glycan epitopes are urgently needed for downstream functional, diagnostic, and other research studies. As described in our paper, our main objective was to first generate and characterize human mAbs to AM/LAM from asymptomatic individuals with Mtb exposure and/or latent infection using single B cell sorting to understand the spectrum of reactivity to glycan epitopes at the human monoclonal level. We show that these mAbs recognize different glycan epitopes, distinct from other anti-AM/LAM mAbs reported. Thus our work adds new and important knowledge as well as new tools for the TB field. We note that it was NOT our objective to do functional investigations with these mAbs, which were beyond the scope of our current studies (lines 70-74; 82-89; 318-322).*
- ii) *We did NOT claim that our mAbs can distinguish latent from active Mtb infection. We did however discuss potential reasons for the narrower AM/LAM epitope specificity of our mAbs generated from asymptomatic individuals with latent Mtb infection compared to broader glycan epitope specificity of the few mAbs generated from B cell cultures of active TB patients (lines 328-330).*
- iii) *Our secondary objective was to generate preliminary data on the value of these mAbs for diagnostic and research tools. While we show mAb binding data in comparison to the well-known anti-LAM murine mAb CS35, this comparison was NOT a main objective but included as a reference. We neither claim that our mAbs have better binding than CS35. CS35, generated from hybridomas of Mycobacterium leprae immunized mice, is an exceptionally high affinity murine mAb with very broad reactivity to many glycan epitopes. However, while the CS35/A194 combo works well with high sensitivity in a urinary LAM capture ELISA, CS35 has problems with aggregation and precipitation over time and cannot be conjugated*

successfully (AuNP or biotin) to achieve high sensitivity in detecting LAM by Lateral Flow Assay (Chatterjee, personal communication). Another issue is that both these mAbs have very broad specificity and thus cross-reactivity with other mycobacteria. Therefore, in addition to the importance for downstream functional investigations, various high affinity anti-AM/LAM mAbs binding to different glycan epitopes need to be generated and evaluated for the detection of LAM in the urine of TB patients. We show that our new mAbs L1AM04 and T1AM09 work well as capture and detection mAbs for the detection of LAM spiked into urine. We now have included the testing of urine samples from two HIV uninfected TB patients and two patients from a TB endemic region who were diagnosed with other respiratory diseases than TB. As shown before for the data with spiked urine, sensitivity and specificity of LAM detection with our new mAbs in these clinical samples was comparable to mAbs CS35/A194 providing preliminary data for larger clinical studies which were beyond the scope of our current studies. This has been clarified throughout the text (e.g. lines 372-385) and a new Table 2 with data on LAM detection in patient samples has been added.

- iv) Additional data on the quantification of mAb binding to virulent and avirulent mycobacterial strains has been added (new Fig. 7 and methods, results, and discussion lines 514-527, 245-255, and 396-400, respectively).

In addition:

Tables 1 & 2 from the first submission were moved to the supplementary section to display only 10 items in the main text (in accordance with the revised manuscript submission file checklist).

Supplementary Figure 1 showing serum antibody titers to AM was added to avoid “data not shown” claims (in accordance with the revised manuscript submission file checklist).

Responses to reviewers' comments point-by-point:

Reviewer #1:

1. While this is the first study to clone anti-AM monoclonal antibodies from LTBI humans using single cell sorting technology, mAbs against Mtb glycolipids LAM and AM were described many times before, including human mAbs.

Response: As we outline in our introduction and discussion, while murine and human mAbs against LAM and AM have been generated before, this is the first report of single B cell sorting as well as using AM to sort for antigen-positive B cells, in addition to using B cells from patients with LTBI (in contrast to TB patients). The focus and novel finding of our described work is the distinct glycan epitope recognition of our anti-AM mAbs. We have clarified this point in the manuscript when discussing (Zimmerman et al 2016) and (Choudhary et al 2018). Lines 76-79.

2. One mAb is used as a positive control in this study and has several orders of magnitude better binding. Moreover, as opposed to the serum of the donors that exhibited interesting anti-mycobacterial activity, none of the antibodies cloned during this study had inhibiting anti-MTB functions, why is this unclear to me.

Response: We did not study the functions of these mAbs because they were not the objective and beyond the scope of the current studies.

3. Overall, though cloned from human origin, the 2 antibodies isolated in this study are not directed against significantly new targets, are not Mtb-inhibiting, and have a weak binding compared to previously described mAbs.

Response: We do not agree with the reviewer on these points. First, we did not study mAb functions, and therefore it cannot be stated that the mAbs are not Mtb-inhibiting. Second, while these two mAbs have weaker binding than the exceptionally high binding of the murine mAb CS35, they still have, for antiglycan mAbs, very high binding. And third, and most importantly, our mAbs recognize glycan epitopes, distinct from other anti-AM/LAM mAbs in the field. This is the first report of a human anti AM/LAM mAb (T1AM09), preferentially binding to the Ara Core epitopes S19/21; other published mAbs require the beta-bond on the terminal arabinose fragments for binding, whereas T1AM09 does not. Similar, in contrast to other anti-AM/LAM mAbs reported to date, L1AM04 is uniquely specific and only binds the AM oligosaccharides with a trimannoside (Man₃) structure, making it distinct from other mannose-specific mAbs. We show that because these mAbs preferentially bind glycan epitopes that other anti-LAM mAbs do not, they likely do not compete with these mAbs (based on glycan array data). Furthermore, they neither compete with each other nor with CS-35 (as shown by BLI in Fig. 4 & S4), and therefore they have utility to be explored in future U-LAM studies.

4. The authors speculate, that due to their LTBI-human origin, and different glycan signature binding, these mAbs can be more useful in discriminating LTBI from active infections, however this is not demonstrated in this study.

Response: We did not make this claim. We did, however, point out that our mAbs, generated from B cells from subjects with LTBI, recognize distinct AM/LAM glycan epitopes compared to two other human mAbs generated from culturing B cells from TB patients, and discussed potential reasons for such distinct glycan recognition between LTBI and TB (3rd paragraph of discussion). While the distinct glycan epitope recognition will be important for the study of Ab functions, we don't not claim it will be useful in discriminating between the states of Mtb infection and TB disease.

5. The authors start with describing their staining, gating and sorting strategy. Can the authors comment why the gating was done using two different strategies? While for T1 the authors gated on CD20/CD7-pos, AM-pos, IgG-pos, for donor L1 the authors gated on CD19-pos/CD3-neg , AM-pos, IgG-pos. If two different strategies were used, a comparison between the two samples cannot be carried out. There is also a clear compensation problem visible on the last T1 gate – the one that was sorted... Figure 1a, right. This might explain why the approximately 80% of the cells that were sorted did not contain functional antibodies when produced. Also, it looks that only 2-3 mAbs produced from each patient were actually specific for the probe. This clearly indicates that the entire procedure was not efficient and there was a lot of noise during the staining and sorting steps.

Response: Our main goal was to generate human mAbs to AM. The protocols for the two sorts were not identical, because our gating strategy has been evolving in effort to obtain higher yields of high-affinity anti-AM mAbs and because there was considerable time between both sorts. Our current protocol implements several fluorescence-minus-one controls and a negative control (mock biotinylated antigen for PE-positive gate). We showed the initial gating strategy for T1AM09 to present the FACS data of the panel of mAbs that we cloned and expressed from our first sort

of subject T1 B cells. We have re-sorted subject T1's AM+ B-cells using the gating strategy of subject L1 (Fig 1b) and identified the same T1AM09 sequence and no other new high-affinity anti-AM/LAM mAbs. We have clarified this further in the text (p. 05, lines 109-112), and added images of the second sort of T1's AM+ B cells in supplemental Fig. S3. We further showed in that other anti-AM mAbs, generated from both our initial and later sorts do bind to AM but at high mAb concentrations. As discussed, and supported by the literature, such low affinity mAbs are to be anticipated among anti-glycan mAbs (p. 06, lines 121-124). We highlight that the isolation of such high affinity anti-AM clones is rare and challenging, regardless of the strategy chosen (screening of in vitro cultures of memory B cells from several TB patients also only yielded two novel mAbs (Choudhary et al 2018)). This has been clarified in the methods (lines 482-487), results (lines 109-115), and discussion (lines 302-303).

6. Based on ELISAs and BLI measurements the antibodies T1AM09 and L1AM04 have weak binding, at least compared to previously described anti-AM mAb CS-35. Nevertheless, the authors repeatedly write that the new mAbs have “high affinity”. Except the BLI, and glycan fingerprints, where previously isolated anti-AM mAb CS-35 was included for comparison, and exhibited 1000-10000-fold better binding, and broader glycan recognition, direct competition or comparison experiments between T1AM09 and L1AM04 and other anti-AM mAbs (CS-35) were not done. Therefore, it is not clear whether the new mAbs bind stronger, or to new sites.

Response: CS35 is a murine mAb that was generated by injecting mice with very high doses of *Mycobacterium leprae*. It is a uniquely high-binding murine mAb that binds to a very broad range of AM/LAM glycan epitopes. While we showed binding to CS35 for comparison, we note that our human mAbs still have, for anti-glycan mAbs, high affinity. Importantly, as outlined in our other responses, the focus of our studies was to show that both of our human mAbs, generated from asymptomatic individuals with latent *Mtb* infection and/or history of *Mtb* exposure, bind to distinct epitopes compared to each other, as well as to other anti-AM/LAM mAbs reported, including CS35. We have outlined this in the text, and clarified further line 174-175. As outlined above, we note that while the CS35/A194 combo works well with high sensitivity in a urinary LAM capture ELISA, CS35 has problems with aggregation and precipitation over time and cannot be conjugated successfully (AuNP or biotin) to achieve high sensitivity in detecting LAM by Lateral Flow Assay (Chatterjee, personal communication). Another issue is both these mAbs have very broad specificity. Therefore, in addition to the importance for downstream functional investigations, broader panels of high affinity anti-AM/LAM mAbs binding to different glycan epitopes need to be generated and evaluated for the detection of LAM in the urine of TB patients. This has been clarified in the discussion, lines 372-385.

7. Interestingly, in Figure 5 using both these mAbs for detection of uLAM was superior to the excising setup of CS-35/A194. However, this was not shown in *Mtb*-infected donor, but rather in a healthy volunteer where LAM was diluted. Also several negative controls are lacking (isotype control, or negative control antigen)

Response: We have added negative control data to the figure and, in response to reviewers 1 and 3, we have included a table on the urine LAM testing results from 4 HIV uninfected patients – two TB patients and two patients from a TB endemic region who were diagnosed with other respiratory diseases than TB. As shown before for the data with spiked urine, sensitivity, and specificity of LAM detection with our new mAbs in these clinical samples was comparable to mAbs

CS35/A194 providing preliminary data for larger clinical studies. We note that larger clinical studies were beyond the scope of our studies and are needed for further evaluation of the value of our mAbs in urinary LAM detection assays. This has been clarified throughout the text (e.g. lines 203-206; 209-214) and a new Table 2 with data on LAM detection in patient samples has been added.

Table 2. Capture and Detection of LAM from urine of subjects with pulmonary TB by Sandwich ELISA.

Diagnosis ^a	Microbiology		U-LAM ^b OD ₄₅₀	ELISA	U-LAM Detection ^c (ng/mL) ^d	GC-MS
	Smear ^e	Culture ^f	CS35 / A194	L1AM04 / T1AM09		
Pulmonary TB	pos	Mtb	1.649	1.531	Positive	25
Pulmonary TB	pos	Mtb	0.894	0.627	Positive	14.3
COPD, Pneumonia	neg	neg	0.233	0.210	Negative	Not detected
Pneumonia, TB unlikely	neg	neg	0.196	0.221	Negative	Not detected

a: all 4 patients were HIV uninfected; b: capture ELISA for the detection of LAM in urine (U-LAM); c: U-LAM detection above cut-off based on background values for healthy control urine for each mAb pair by capture ELISA (OD₄₅₀ 0.259 for CS35/A194 mAb pair and 0.335 for L1AM04/T1AM09 mAb pair, respectively); d: quantification of U-LAM by gas chromatography-mass spectrometry (GS-MS) D-Arabinose detection method; e: sputum smear microscopy for acid fast bacilli (AFB); (f) mycobacterial culture of sputum

8. In Figure 6 the authors write that there is “strong binding” to various strains of Mtb, based on microscopy. However, the degree of staining signal is not clear from the figure. For example, the authors write that there is binding to M. abcessus, whereas the staining looks very weak. What % of the bacteria is bound by the mAbs? How is it compared to the anti-AM mAb CS-35. The same is true for Figure 7.

Response: We assessed mAb binding of bacteria at the single bacilli level and at 20x magnification to capture a large area with many single bacilli (not the clumps) in the field of view. We rely on using the same culture conditions, staining conditions, and image analysis settings to relatively compare the mAbs binding to the different strains. It is challenging to quantitate the % of bacteria positive for mAb staining for multiple reasons: software cannot distinguish between

single bacilli and debris at 20x, and confocal microscopy cannot detect the unstained cells (phase contrast). When magnification is increased, it isn't feasible to capture the true positive binding of many single bacilli. To complement the fluorescent microscopy experiments and add quantitative binding data, we have added a new figure with whole cell ELISA data (Fig 7). OD and protein concentration of each culture were used to normalize the number of bacteria coated on to the ELISA plate. This has been described in the methods (lines 514-527) and references are provided, and text to this effect has been added to the results (lines 245-255) and discussion (lines 396-400).

Figure 7. T1AM09 and L1AM04 show distinct binding to mycobacterial strains by whole cell ELISA (mycobacteria were grown without detergent to preserve the capsule). Binding of human IgG1 mAbs (50 µg/mL) and polyclonal serum IgG (1:20 dilution) to virulent laboratory (H37Rv and Erdman) and clinical strains (CDC1551 and Beijing) of *Mtb*, avirulent strains of the *Mtb* complex group (H37Ra and BCG Pasteur) and nontuberculous mycobacteria (*M. avium* and *M. abscessus*) assessed at optical density (OD) of 450 nm for (a) T1AM09; (b) L1AM04; (c) Isotype matched negative control IgG1 mAb to a flavivirus; (d) Subject T1 serum; and (e) Subject L1 serum. Data shown are representative from two separate experiments.

9. In the Discussion section the authors discuss germline restriction. They write that one of their findings is that Vh3 is overrepresented in anti-Mtb mAbs (similarly to COVID-19 and Malaria). However, Vh3 is the most common Vh within the human Vh gene repertoire, therefore was this claim based on normalization to the individual frequency on each Vh within the Vh gene repertoire? This analysis should be included in the figures, or this claim cannot be made.

Response: We agree with the Reviewer, have clarified that in humans the VH3 germline accounts for about 35-40% of the antibody repertoire, and have modified this section in the discussion lines 343-346.

10. Also, the claim that Vh1-2 antibodies in HIV-1 “are associated with broad polyclonal reactivity to lipopolysaccharides” is completely untrue. While some Vh1-2 mAbs maybe bind sugars on top of the heavily glycosylated HIV-1 (which are in a completely different configuration as the authors themselves note from the mycobacteria LAM and AM), the most potent anti-HIV-1 CD4 bindings site Vh1-2 arising neutralizing antibodies are NOT glycan dependent (VRC01, 3BNC117, 3BNC60 and more)...

Response: The reviewer misinterpreted the sentence, so we revised our statement for better clarity lines 349-352. Our highest affinity mAb, T1AM09, is from the VH1-2 germline. This germline is associated with the class of broadly neutralizing antibodies (bNAbs) to the HIV-1 CD4-binding site, and while these bNAbs are not all glycan dependent, they are reported to develop flexible complementarity-determining regions (CDRs) to tolerate glycan shields on these viruses (reviewed in (51)). VH1-2 germline antibodies are further reported with a broad reactivity to lipopolysaccharides from several species of gram-negative bacteria (52).

11. Figure 2 – no negative or positive control antibody is shown. Also figure fonts should be the same size.

Response: We have added the BSA control to the figure and have corrected fronts to same size.

12. In Supplementary FACS plots, axis labels should be clearly indicated.

Response: This has been done.

13. Glycan fingerprint from Supplementary Figure 3 should be included in main figure 3 for comparison

Response: Because the CS-35 glycan fingerprint was previously published (4) we provide CS-35 binding data in the supplemental to present consistency of our binding assays with published work (4). CS-35 is not a true positive control because it is a murine antibody and different secondary antibodies were used for microarray and ELISA; thus, magnitude of binding (MFI or OD₄₅₀) is not exactly comparable.

14. Figure 6 and Supplementary figure 4 – there is no scale

Response: We have added a 10 um scale bar to the figures.

Reviewer #2:

15. In the Discussion section, the authors state that VH3 germline genes are overrepresented for human mAbs to MtbAM. I think this statement might need to be clarified or revised. VH3 is one of the most common VH subgroups in humans - from one paper (Arnaout Plos one 2011), VH3 is about 35-40% of all human VH. So, a random sample of human antibodies might have roughly the same proportion of VH3 as their mAbs. Therefore, I'm not sure I would call them overrepresented, or perhaps they might clarify.

***Response:** We agree with the Reviewer, have clarified that in humans the VH3 germline accounts for about 35-40% of the antibody repertoire, and have edited these comments accordingly (lines 343-346.*

16. Figure 3. For the glycan structures, I would probably add a description of what the different bond direction mean. I suspect most non-glyco people will not understand that the direction means something. Maybe state in the legend that all bonds are alpha unless listed in the figure as beta.

***Response:** We added the orientation of the bonds into the figure legend.*

17. As a reader, I would like the full sequences of all the antibodies to be listed in the Supp. What sort of amino acid mutations occur? where are they located? Also, it would enable others to evaluate and use the antibodies. Of course, there may be IP issues. I will leave it to the Editors to decide if the sequences are required or not.

***Response:** We have provided Gene accession numbers at GenBank Record (<https://www.ncbi.nlm.nih.gov/genbank/samplerecord/>). This information has been added to the Data Availability Section, lines 598-601.*

The accession numbers for the nucleotide and protein sequences for variable regions of T1AM09 and L1AM04 are GenBank: MZ600149 - MZ600152. Glycan array data were deposited in the NCBI Gene Expression Omnibus (GEO) database with accession number GSE180517.

18. Experimental details: In many places throughout the Methods section, the sources of chemicals/antibodies/proteins, working concentrations, and buffers are not specified. For example, several places list a dilution of 1:200 but the buffer is not listed. If BSA is used in the buffers, list the source, quality, and working concentration. As a second example, the buffer and concentration of the SA-Cy5 secondary for the glycan microarray assay are not listed. Please add this information throughout the Methods section.

***Response:** This information has been provided throughout the methods section.*

Reviewer #3:

19. Lines 40-42: While figure 6 does show strong detection of the virulent Mtb strains and BCG with either T1AM09 or L1AM04, weak detection of M. abscessus only by T1AM09, and neither causing H37Ra or M. avium to fluoresce red, it is unclear if the lack of or limited detection of the nontuberculous strains is due to reduced clumping of these strains, reduced production of the AM targets of these mAbs, or a microscope resolution or intensity setting issue. The case for detection could be made stronger by including some panels in which the strains also expressed GFP. AECOM has renowned Mtb investigators who have GFP--encoding plasmids for use in mycobacteria and may even have these species already engineered to express GFP.

***Response:** While constitutively GFP expressing mycobacteria could be useful, many of the strains that we investigated for mAb binding are not readily available and would need to be designed, cloned, transformed, and optimized (especially the NTM strains). This addition will also not confirm/quantitate the differential binding and fluorescent intensity between the mAbs binding to different mycobacteria strains. To complement the fluorescent microscopy and add quantitative data, we have added results of whole cell ELISA with the mycobacterial strains (Fig 7). By ELISA, these data confirm reduced binding of T1AM09 and L1AM04 to M. avium and M. abscessus compared to positive binding to M. avium and M. abscessus detected with sera from subjects T1 and L1, highlighting the role of glycan epitope specificity in the binding of mAbs to NTMs. OD and protein concentration of each culture were used to normalize the number of bacteria coated on to the ELISA plate (1, 2). This has been described in the methods and references are provided, and text to this effect has been added to the results and discussion (methods, results, and discussion lines 514-527, 245-255, and 396-400, respectively).*

Figure 7. T1AM09 and L1AM04 show distinct binding to mycobacterial strains by whole cell ELISA (mycobacteria were grown without detergent to preserve the capsule). Binding of human IgG1 mAbs (50 μ g/mL) and polyclonal serum IgG (1:20 dilution) to virulent laboratory (H37Rv and Erdman) and clinical strains (CDC1551 and Beijing) of *Mtb*, avirulent strains of the *Mtb* complex group (H37Ra and BCG Pasteur) and nontuberculous mycobacteria (*M. avium* and *M. abscessus*) assessed at optical density (OD) of 450 nm for **(a)** T1AM09; **(b)** L1AM04; **(c)** Isotype matched negative control IgG1 mAb to a flavivirus; **(d)** Subject T1 serum; and **(e)** Subject L1 serum. Data shown are representative from two separate experiments.

20. Lines 261-2: lack of positive staining by AFB outside of the inflammatory region in panel? Panel F appears to have a large number of infiltrating cells. Please clarify what is meant by outside the inflammatory region.

***Response:** Pathologists noted that AFB staining was only detected in major areas of tissue inflammation where clumps of bacilli were located. The region (Fig. 8F) is approximately the same section of the lung as shown in the other panels of Fig. 8 where positive mAb staining was detected. This has been clarified in the figure legend.*

21. (old) Figure 7. Why are the uninfected mouse tissues (panels g-i) at a much lower resolution than the Mtb-infected tissues? For apples-to-apples comparison of the extent of background staining, similar resolution should be used. It would be especially interesting to know if the diffuse tan staining in the mAb is the result of the presence of the bacteria and how it compares to the isotype control.

***Response:** We presented a lower magnification for unaffected murine tissue (g-i) to show that non-specific staining was not detected in a larger area but have changed the magnification for the controls to be consistent throughout the panels (Figure 8). The diffuse tan staining was not present in any of the uninfected tissue IHC staining.*

Figure 8: T1AM09 and L1AM04 detect extra- and intracellular *Mtb* and LAM in lung tissues of *Mtb*-infected mice. Histology and immunohistochemistry of *Mtb* (CDC1551) infected murine lung (scale bar 100 μ m) showing intra- and extracellular staining of *Mtb* by (a) T1AM09; (b) L1AM04; and (c) Staining for acid-fast bacilli (AFB); (d) Staining of intracellular LAM in *Mtb* (CDC1551) infected lungs (arrows indicate LAM within macrophages) by T1AM09; (e) Staining of intracellular LAM and single bacilli in *Mtb* Erdman infected lungs (arrow indicates LAM within macrophages) by L1AM04; and (f) lack of positive AFB staining of *Mtb* (Erdman) in approximately the same *Mtb* infected lung section as shown in the other figure panels. Overall lack of staining of non-infected murine tissue (scale bar 100 μ m) by (g) T1AM09; and (h) L1AM04. (i) Lack of staining of *Mtb* (CDC1551) infected lung tissue by isotype matched control mAb to a flavivirus (scale bar 100 μ m). All mAbs were tested at 2 μ g/mL.

22. Lines 78-81: Please clarify this sentence

“Among these, the high affinity and best characterized mAb A194 recognizes a range of AM oligosaccharide motifs (4). We here used single B cell sorting from peripheral blood mononuclear cells (PBMC) of asymptomatic *Mtb* exposed or infected individuals with high serum anti-AM IgG titers, protective polyclonal anti-AM IgG functions, and diverse glycan epitope binding (5).”

Response: We have clarified that “Among these, the best characterized human mAb A194 has high affinity and recognizes a range of AM oligosaccharide motifs sharing the uncapped Ara4 and Ara6 epitopes commonly recognized by murine mAbs (17). We here used AM-specific single B cell sorting to develop a targeted approach for generating mAbs from asymptomatic *Mtb* exposed and/or latently infected individuals. The sera from these individuals had high anti-AM IgG titers, protective polyclonal anti-AM IgG functions, and diverse glycan epitope binding (15)” (lines 76 – 82).

23. Line 157: glycopeptidolipids?

Response: Glycopeptidolipids (new line 158) is correct. These are antigens found in NTMs of the *Mycobacterium avium* complex (MAC), especially in *M. avium* and *M. intracellulare* – see ref (6).

24. Line 225: somewhere in the legend to figure 5 it should be stated that this is a sandwich ELISA.

Response: This has been stated.

25. Line 240: Indicate the magnification used in this figure.

Response: This has been done.

26. Lines 302-304: Please clarify this sentence. Present in all *Mtb* and most other pathogenic mycobacterial strains, ManLAM plays an important role in *Mtb* phagocytosis, intracellular trafficking, and the impact of immune responses, assuring *Mtb*'s survival within infected host cells (reviewed in (7)).

Response: We have clarified the sentence (lines 315-318) to “ManLAM is present in *Mtb* and most other pathogenic mycobacterial strains and facilitates *Mtb* phagocytosis and intracellular trafficking, thereby contributing to *Mtb*'s survival within infected host cells (reviewed in (7)).”

27. Line 350: For clinical relevance, the antibody pair must be able to detect AM or LAM epitopes that remain intact. While the results with the LAM-spiked urine look great in figure 5, it was disappointing that no comparisons of the L1AM04/T1AM09 pair to CS-

35/A194 were performed with urine from TB-positive humans or from urine collected from Mtb-infected animals if human samples are not available.

Response: We have tested L1AM04/T1AM09 with urine from two TB patients and two patients with respiratory diseases other than TB showing similar sensitivity and specificity compared to CS-35/A194 (new Table 2).

Table 2. Capture and Detection of LAM from urine of subjects with pulmonary TB by Sandwich ELISA.

Diagnosis ^a	Microbiology		U-LAM ^b OD ₄₅₀	ELISA	U-LAM Detection ^c (ng/mL) ^d	GC-MS
	Smear ^e	Culture ^f	CS35 A194	/ L1AM04 / T1AM09		
Pulmonary TB	pos	Mtb	1.649	1.531	Positive	25
Pulmonary TB	pos	Mtb	0.894	0.627	Positive	14.3
COPD, Pneumonia	neg	neg	0.233	0.210	Negative	Not detected
Pneumonia, TB unlikely	neg	neg	0.196	0.221	Negative	Not detected

a: all 4 patients were HIV uninfected; b: capture ELISA for the detection of LAM in urine (U-LAM); c: U-LAM detection above cut-off based on background values for healthy control urine for each mAb pair by capture ELISA (OD₄₅₀ 0.259 for CS35/A194 mAb pair and 0.335 for L1AM04/T1AM09 mAb pair, respectively); d: quantification of U-LAM by gas chromatography-mass spectrometry (GS-MS) D-Arabinose detection method; e: sputum smear microscopy for acid fast bacilli (AFB); (f) mycobacterial culture of sputum

28. Line 374: extraceulluar?

Response: Extracellular - typo fixed (now line 407)

29. Line 412: which minimal medium was used to culture the strains?

To allow for mycobacterial capsule formation, the pre-culture strains were inoculated in minimal medium (MM) without detergent at 37°C for 3 weeks (as previously described, (8)). Minimal medium consisted of 1 g/l KH₂PO₄, 2.5 g/l Na₂HPO₄, 0.5 g/l asparagine, 50 mg/l ferric ammonium citrate, 0.5 g/l MgSO₄ × 7 H₂O, 0.5 mg/l CaCl₂, 0.1 mg/l ZnSO₄, with or without 0.05% tyloxapol (v/v), containing 0.1% (v/v) glycerol, pH 7.0. This information has been added to the methods (lines 466-469).

30. Supplementary Figure 3: Please label to top of the panel A drawing as done in Figures 3 A and B.

Response: This has been done; now Supplementary Figure 4.

31. Supplementary Figure 5: Is it possible to conclude that this staining method lack sensitivity based on the images in this figure alone? What is the positive control for Fite's method? Why is only the pink color with the arrow in panel B the only positive staining? What is the pink staining above it?

Response: Our pathologist noted the clumps of positive staining are multifocal clusters of nonspecific staining and noted the positive bacilli were only visible in area pointed at by the arrow. The positive control tissue for Fite's is from banked Mtb-infected murine lung sections. There are numerous reasons why the control tissue reacted with Fite's compared to our experimental tissue: different infection doses, sectioning, and metabolic state of the Mtb at the time of fixation could have all impacted Acid-Fast Bacilli positive staining (9, 10). This has been clarified in the text (lines 268-270; 404-406) and the supplemental figure 6 legend.

Select references:

1. Meyers PR, Bourn WR, Steyn LM, van Helden PD, Beyers AD, Brown GD. Novel method for rapid measurement of growth of mycobacteria in detergent-free media. *J Clin Microbiol.* 1998;36(9):2752-4. Epub 1998/08/15. doi: 10.1128/jcm.36.9.2752-2754.1998. PubMed PMID: 9705430; PMCID: PMC105200.
2. Schwebach JR, Casadevall A, Schneerson R, Dai Z, Wang X, Robbins JB, Glatman-Freedman A. Expression of a Mycobacterium tuberculosis arabinomannan antigen in vitro and in vivo. *Infect Immun.* 2001;69(9):5671-8. doi: 10.1128/IAI.69.9.5671-5678.2001. PubMed PMID: 11500443.
3. Zimmermann N, Thormann V, Hu B, Köhler AB, Imai-Matsushima A, Loch C, Arnett E, Schlesinger LS, Zoller T, Schürmann M, Kaufmann SH, Wardemann H. Human isotype-dependent inhibitory antibody responses against *Mycobacterium tuberculosis*. *EMBO Molecular Medicine.* 2016;8(11):1325-39. doi: 10.15252/emmm.201606330; PMCID: PMC5090662.
4. Choudhary A, Patel D, Honnen W, Lai Z, Prattipati RS, Zheng RB, Hsueh Y-C, Gennaro ML, Lardizabal A, Restrepo BI, Garcia-Viveros M, Joe M, Bai Y, Shen K, Sahloul K, Spencer JS, Chatterjee D, Broger T, Lowary TL, Pinter A. Characterization of the Antigenic Heterogeneity of Lipoarabinomannan, the Major Surface Glycolipid of *Mycobacterium tuberculosis*, and Complexity of Antibody Specificities toward This Antigen. *The Journal of Immunology.* 2018. doi: 10.4049/jimmunol.1701673; PMCID: PMC5911930.
5. Chen T, Blanc C, Liu Y, Ishida E, Singer S, Xu J, Joe M, Jenny-Avital ER, Chan J, Lowary TL, Achkar JM. Capsular glycan recognition provides antibody-mediated immunity against tuberculosis. *The Journal of clinical investigation.* 2020. Epub 2020/01/15. doi: 10.1172/jci128459. PubMed PMID: 31935198.
6. Zheng RB, Jégouzo SAF, Joe M, Bai Y, Tran HA, Shen K, Saupe J, Xia L, Ahmed MF, Liu YH, Patil PS, Tripathi A, Hung SC, Taylor ME, Lowary TL, Drickamer K. Insights into Interactions of Mycobacteria with the Host Innate Immune System from a Novel Array of Synthetic

- Mycobacterial Glycans. ACS Chem Biol. 2017;12(12):2990-3002. Epub 2017/10/20. doi: 10.1021/acscchembio.7b00797. PubMed PMID: 29048873; PMCID: PMC5735379.
7. Turner J, Torrelles JB. Mannose-capped lipoarabinomannan in Mycobacterium tuberculosis pathogenesis. Pathog Dis. 2018;76(4). Epub 2018/05/04. doi: 10.1093/femspd/fty026. PubMed PMID: 29722821; PMCID: PMC5930247.
 8. Prados-Rosales R, Baena A, Martinez LR, Luque-Garcia J, Kalscheuer R, Veeraraghavan U, Camara C, Nosanchuk JD, Besra GS, Chen B, Jimenez J, Glatman-Freedman A, Jacobs WR, Jr., Porcelli SA, Casadevall A. Mycobacteria release active membrane vesicles that modulate immune responses in a TLR2-dependent manner in mice. The Journal of clinical investigation. 2011;121(4):1471-83. Epub 2011/03/03. doi: 10.1172/jci44261. PubMed PMID: 21364279; PMCID: PMC3069770.
 9. Farver CF, Jagirdar J. 11 - Mycobacterial Diseases. In: Zander DS, Farver CF, editors. Pulmonary Pathology (Second Edition). Philadelphia: Content Repository Only!; 2018. p. 201-16.
 10. Vilchèze C, Kremer L. Acid-Fast Positive and Acid-Fast Negative Mycobacterium tuberculosis: The Koch Paradox. Microbiol Spectr. 2017;5(2). Epub 2017/03/25. doi: 10.1128/microbiolspec.TBTB2-0003-2015. PubMed PMID: 28337966.

REVIEWERS' COMMENTS:

Reviewer #1 (Remarks to the Author):

The authors significantly revised the manuscript and addressed many of my comments. I am convinced that this work is important and deserves to be published.

I have two minor points

-New Figure 7 – a binding of a control human serum to the same bacterial strains (in addition to L1 and T1 sera) needs to be shown. Control serum is the equivalent of the Isotype negative control for the sera.

-Recently there have been several MTB human monoclonal antibodies generated using a similar approach (while directed against different targets) and these studies worth mentioning.

Reviewer #2 (Remarks to the Author):

the authors have addressed all of my comments and concerns.

Reviewer #3 (Remarks to the Author):

The authors have made significant improvements to the manuscript. I only have minor grammatical suggestions.

Line 251: Please delete "did".

Please italicize "Mtb" in Table 2 and on Lines 235.

Line 317: Please change "Mtb's" to "Mtb" and italicize it

Line 322: check spelling/grammar: is "trimannoside-capped" correct?

Line 324: add hyphen between mannose and capped

Line 346: suggest rewording the last half of the sentence as follow: these VH3 mAbs had affinities too low to be further characterized

Line 377: delete the comma?

Line 382: This sentence is cumbersome. Either indicate the unique features of the antibodies that support they can be combined with other high-affinity anti-LAM mAbs in U-LAM detection assays or replace ", due to their unique features, could" with "have potential to be"

Line 384: valube is misspelled

Line 457: suggest changing "one with...." to "one with pneumonia, the other with pneumonia and chronic obstructive lung disease."

We thank the reviewers for their time, comments, and input. We are happy to hear that they felt we addressed their comments, significantly improved our resubmitted paper, and that our work is important and suitable for publication in *Communications Biology*. Below are our responses to their last remaining minor comments (changes in response to reviewers' comments are highlighted in final revision). We also added a section on authors' contributions at the end.

Reviewer #1:

The authors significantly revised the manuscript and addressed many of my comments. I am convinced that this work is important and deserves to be published. I have two minor points:

New Figure 7 – a binding of a control human serum to the same bacterial strains (in addition to L1 and T1 sera) needs to be shown. Control serum is the equivalent of the Isotype negative control for the sera.

Response: We do not agree with this request. Our main objective was the characterization of human anti-AM/LAM mAbs generated from asymptomatic M. tuberculosis exposed/infected individuals and show their value for TB diagnostics and research. In response to reviewers' comments, we performed additional work (whole cell ELISA) and added data on the quantification of our two mAbs binding to virulent and avirulent mycobacterial strains (new Fig. 7). In the absence of known positive controls reacting with all Mtb and NTM strains, L1 and T1 sera were included as positive controls. However, we included an isotype matched negative control mAb rather than serum, because it is the optimal negative control for the assessment of IgG1 mAb binding, especially as most sera, even from M. tuberculosis uninfected individuals, contain some amount of polyclonal IgG against AM/LAM (published by us in ref 11, Yu, et al. Comparative evaluation of profiles of antibodies to mycobacterial capsular polysaccharides in tuberculosis patients and controls stratified by HIV status. Clinical and vaccine immunology, 2012) Positive and negative controls for whole cell ELISA were/are outlined in the methods and "positive controls" has been added to the legend of Fig. 7.

Recently there have been several MTB human monoclonal antibodies generated using a similar approach (while directed against different targets) and these studies are worth mentioning.

Response: We added a sentence to the discussion that "For TB, two recent studies used antigen-specific memory B cell sorting to generate mAbs to the Mtb proteins heparin-binding hemagglutinin (HBHA) and phosphate-binding lipoprotein (PstS1)." and included the references (lines 302 – 304).

Reviewer #2:

The authors have addressed all of my comments and concerns.

Reviewer #3:

The authors have made significant improvements to the manuscript. I only have minor grammatical suggestions.

Line 251: Please delete "did".

Done.

Please italicize “Mtb” in Table 2 and on Lines 235.

Done.

Line 317: Please change “Mtb’s” to “Mtb” and italicize it.

Done.

Line 322: check spelling/grammer: is “trimannoside-capped” correct?

Correct.

Line 324: add hyphen between mannose and capped.

Added.

Line 346: suggest rewording the last half of the sentence as follow: these VH3 mAbs had affinities too low to be further characterized.

Reworded accordingly.

Line 377: delete the comma?

Comma deleted.

Line 382: This sentence is cumbersome. Either indicate the unique features of the antibodies that support they can be combined with other high-affinity anti-LAM mAbs in U-LAM detection assays or replace “, due to their unique features, could” with “have potential to be”

Corrected.

Line 384: valuabe is misspelled

Corrected.

Line 457: suggest changing “one with....” to “one with pneumonia, the other with pneumonia and chronic obstructive lung disease.”

Changed accordingly.